# Accessing ultrastable glass via a bulk transformation

Hengtong Bu [1], Hengwei Luan [1,2,3], Jingyi Kang[1], Jili Jia[1], Wenhui Guo[1], Yunshuai Su[1], Huaping Ding[4], Hsiang-Shun Chang[1], Ranbin Wang[1], You Wu[1], Lingxiang Shi [1], Pan Gong [4], Qiaoshi Zeng [5,6], Yang Shao [1] ✉ & Kefu Yao[1] ✉

As a medium to understand the nature of glass transition, ultrastable glasses have garnered increasing attention for their significance in fundamental science and technological applications. Most studies have produced ultrastable glasses through a surface-controlled process using physical vapor deposition. Here, we demonstrate an approach to accessing ultrastable glasses via the glass-to-glass transition, a bulk transformation that is inherently free from size constraints and anisotropy. The resulting ultrastable glass exhibits a significantly enhanced density (improved by 2.3%), along with high thermodynamic, kinetic, and mechanical stability. Furthermore, we propose that this method of accessing ultrastable glasses is general for metallic glasses, based on the examination of the competitive relationship between the glass-to-glass transition and crystallization. This strategy is expected to facilitate the proliferation of the ultrastable glass family, helping to resolve the instability issues of glass materials and devices and deepen our understanding of glasses and the glass transition.

Glasses, as one of the most interesting materials in solid-state physics[1], typically form when the liquids undergo sufficiently rapid cooling, effectively bypassing the crystallization process. As the temperature decreases, the motion of the molecules/atoms gradually decelerates until it is no longer able to sample equilibrium configurations of the supercooled liquid on the laboratory timescale, where the glass forms[2]. This kinetic bottleneck obscures the nature of glass transition and results in loosely-packed metastable glasses that lack sufficient stability. Therefore, accessing highly stable glasses, or even the so-called ideal glass, is of great interest not only for addressing those long-puzzled fundamental questions, such as whether there is a thermodynamic glass transition underlying the observed glass formation[3–6], but also for the technological applications, specifically obtaining denser glasses with high stability[7].

Natural aging is a conventional way to transit liquid-cooled glasses to lower energy states. Nevertheless, due to the limited mobility of molecules/atoms at room temperature (RT) and the logarithmically slowing aging process over time[8], obtaining ultrastable glasses through natural aging generally demands thousands to millions of years[9,10]. In the search for ultrastable glasses, in 2007 it was noticed that the ultrastable organic molecular glass films with lower enthalpies and higher densities could be obtained via physical vapor deposition (PVD)[4,11] by adjusting the substrate temperature and deposition rate. Thanks to the rapid surface atomic diffusion[12], the PVD method takes a few tens of hours to attain an ultrastable glass film with thickness in the order of tens to hundreds of nanometers. Thereafter, many ultrastable glass films, including organic[13–15], metallic[16–18], polymer[19–21], and chalcogenide[22,23] glasses, were prepared using the PVD method. To the

[1]School of Materials Science and Engineering, Tsinghua University, Beijing 100084, China. [2]Department of Mechanical Engineering, City University of Hong Kong, Hong Kong 999077, China. [3]City University of Hong Kong Matter Science Research Institute (Futian), No. 3, Binglang Road, Futian District, Shenzhen 518045, China. [4]State Key Laboratory of Materials Processing and Die & Mould Technology, School of Materials Science and Engineering, Huazhong University of Science and Technology, Wuhan 430074, China. [5]Center for High Pressure Science and Technology Advanced Research, Shanghai 201203, China. [6]Shanghai Key Laboratory of Material Frontiers Research in Extreme Environments (MFree), Institute for Shanghai Advanced Research in Physical Sciences (SHARPS), Shanghai 201203, China. ✉e-mail: shaoyang@tsinghua.edu.cn; kfyao@mail.tsinghua.edu.cn

best of our knowledge, the PVD method is the only way to produce different kinds of ultrastable glasses on the laboratory timescale so far.

Despite the great progress advanced by the PVD method, the sample size is still very limited owing to the intrinsic slow deposition rate of PVD. Due to the surface-mediated process, only two-dimensional glassy films are now accessible and sometimes markedly anisotropic[24,25]. Therefore, it is still a challenge to obtain ultrastable glasses through a bulk process, if possible, free from substrates and with no anisotropy.

Current researches on ultrastable glasses are predominantly focused on the field of organic glasses. However, the directional covalent bonds will bring more kinetic barriers that could retard the atoms/molecules to sample the equilibrium configuration, resulting in a need for high substrate temperatures and low deposition rates in PVD. Whereas, metallic glasses, owing to their simple structure and the non-directional metallic bonds, are more easily to surmount the kinetic barriers and thereby access ultrastable states. According to some sporadic reports in recent years, an anomalous exothermic peak prior to crystallization of some metallic glasses[26–32] during the differential scanning calorimetry (DSC) heating process, sheds some light. After annealing at the end temperature of the exothermic peak, the resulting sample was found to be still amorphous without (nano-)crystallization or phase separation, *i.e.*, a glass-to-glass transition (GGT) occurred. Due to the exothermic event, the resulting glass evidently exhibits a lower enthalpy state. Additionally, an increase in the onset glass transition temperature ($T_g$) was also noticed[27,28,30]. Those recent reports on metallic glasses induced by GGT (sometimes referred to as liquid-to-liquid transition), drop a hint that a new route to access ultrastable glass state may be possible through a bulk transformation, rendering the sample-size limitation no longer a concern. However, those studies mainly focused on the transition itself but overlooked the nature of the newly-formed glass. Hence, it is still elusive whether GGT-induced glasses are undoubtedly ultrastable, particularly in the absence of density variation measurements, and further, whether GGT is a general route for metallic glasses to access ultrastable states.

In this paper, by surveying the GGT in Ti-Zr-Cu-Ni-Be metallic glasses, we provide evidence that the resulting glass induced by GGT is an ultrastable glass with a 2.3% increase in density, along with excellent thermodynamic, kinetic, and mechanical stability, proceeding 75% toward the ideal glass state. More importantly, we propose that GGT could be a general route to obtain ultrastable glasses, at least in metallic glasses, through a bulk transformation. Compared with the vapor-deposited ultrastable glasses generated by a surface-controlled process, the GGT approach offers advantages of high efficiency, no need for substrates, no size limitations, and no anisotropy. This paves the way for the discovery of more ultrastable glasses, contributing to deepening our understanding of the nature of glass and glass transition.

## Results

### A glass-to-glass transition in TiZrCuNiBe metallic glass

The DSC curves of the equal-atomic TiZrCuNiBe glassy ribbons are shown in Fig. 1a. After the glass transition, there appear three exothermic peaks for the as-prepared sample, which were recognized as crystallization or crystal transformation[33,34]. After heating to the end of the first post-$T_g$ peak ($T_{HT}$ = 781 K) and immediately cooling down, the heat-treated sample was obtained, and some intriguing things were noticed: the newly-formed sample still exhibits a glass transition ($T_{g\text{-}HT}$ = 642 K); meanwhile the first post-$T_g$ peak completely disappears while the other two remain (without enthalpy change). The X-ray diffraction (XRD) spectra of the as-prepared and heat-treated samples are shown in Fig. 1b. Broad rather than sharp diffraction peaks indicate the amorphous structure of both the two samples. High-resolution transmission electron microscopy (HRTEM) and selected

area electron diffraction (SAED) further confirm that no nano-crystals form during the first post-$T_g$ exothermic event and the heat-treated sample is fully amorphous (Fig. 1e, g, more images in Supplementary Fig. 1). Moreover, no compositional inhomogeneity was observed in the high-angle annular dark-field (HAADF) images sensitive to element distribution for both the as-prepared and heat-treated samples (Fig. 1f, h). Atom probe tomography (APT) analyses were also carried out and uniform element distribution in sub-nanometer scales was observed before and after the exothermic event (see Supplementary Fig. 2), further excluding the possibility of decomposition. The results above indicate that the heat-treated sample is still a monolithic amorphous phase and the first post-$T_g$ exothermic event is in fact a glass-to-glass transition rather than (nano-)crystallization or phase separation. Afterwards, a two-stage crystallization process occurs, manifested as the second and third post-$T_g$ exothermic peaks[33,34].

### Structural ordering

Synchrotron X-ray diffraction was conducted to investigate the structural changes induced by GGT. Figure 1c shows the total scattering factor $S(Q)$ of the as-prepared and heat-treated TiZrCuNiBe glassy ribbons, and the inset is a close-up of the first sharp diffraction peaks (FSDPs). The peak position ($Q_1$) and integral width ($W$) of the FSDPs for the as-prepared and heat-treated samples were calculated and listed in Supplementary Table 1. After the GGT, the peak position of the FSDP shifts to a higher $Q$ by 0.38%, indicating a smaller average atomic volume/spacing; the integral width of the FSDP decreases by 17.5%, denoting a more ordered structure induced by the GGT.

The reduced pair-distribution functions $G(r)$ of the as-prepared and heat-treated samples are shown in Fig. 1d. The shape of the first peak makes a great difference after heat treatment, indicating a great change in the short-range order. However, due to the presence of 15 different kinds of atom pairs, it is of great difficulty to quantitatively analyze the local structural evolution. Qualitatively, there appears to be an increase in the proportion of atomic pairs exhibiting negative mixing enthalpy as a result of the GGT (see Supplementary Fig. 3), consistent with previous results[35]. In addition, $G(r)$ peaks behave significantly different damping rates, manifesting that the medium-range order also goes through a great change during the glass-to-glass transition. Theoretically, the peaks in $G(r)$ for a perfect crystal will extend infinitely; while for glasses lacking long-range order, the peaks will damp off very quickly and end up at a point where there are no more correlations. To measure how quickly the peaks damp as a function of $r$ for the as-prepared and heat-treated samples, an exponential decay function $h(r)=A\exp(-r/\xi)$ was applied to fit the peak heights and peak positions within 16 Å (inset of Fig. 1d, detailed fitting results can be seen in Supplementary Table 2; note that the second peaks were skipped to obtain a better fitting)[36–38], where the fitting parameter $\xi$ is a measure of the decay rate of $h(r)$ and $A$ refers to the amplitude. The heat-treated sample has a much larger $\xi$, indicating a slower damping rate and therefore the ordering ends at a farther point.

To quantify how far the structural order can be extended, we define an ordering-range (OR) parameter as

$$\textbf{OR} = \max\{r|h(r) \geq 3I_{NS}\} \tag{1}$$

where $h(r)$ refers to the above fitting function, and $I_{NS}$ refers to the average intensity (the absolute value of heights) of the noise signals extracted from the range of 30 – 40 Å in $G(r)$. Note that this definition assumes that signals exceeding three times the average intensity of the noise signals are considered as effective structural ordering information. Then, the ordering-range parameters for the as-prepared and heat-treated samples are calculated to be 18.3 Å and 25.8 Å, respectively (Table 1). It is found that after the GGT, the ordering-range extends to a longer distance, increasing by 41.2%. The results suggest a more stable structure with possible denser packing.

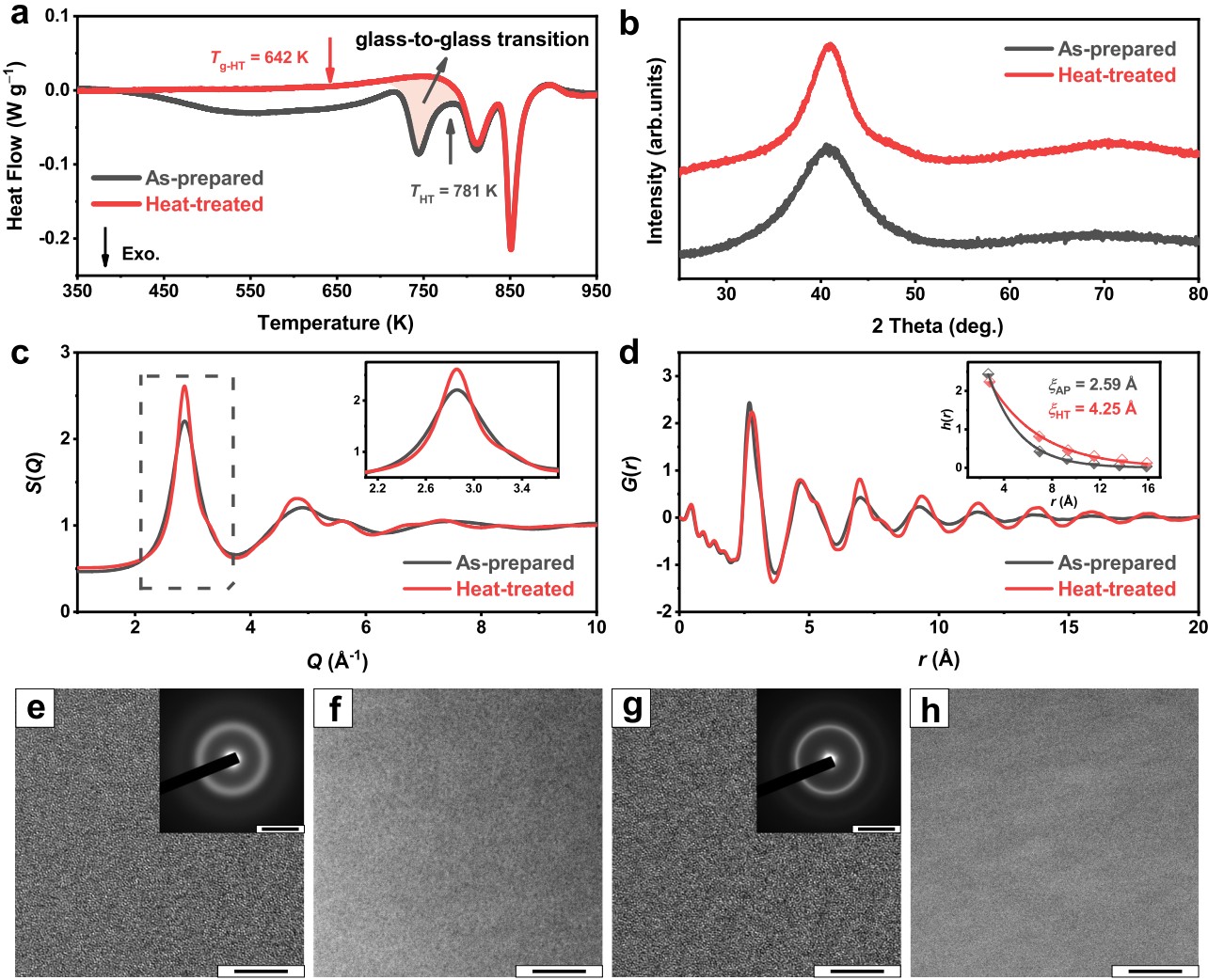

**Fig. 1 | The glass-to-glass transition (GGT) in TiZrCuNiBe metallic glass.**
**a** Differential scanning calorimetry (DSC) curves of the as-prepared and heat-treated samples with a heating rate of 10 K min⁻¹. When heating, the as-prepared sample undergoes relaxation, GGT (the orange-shaded area), and crystallization in sequence. The GGT enthalpy is 45.6% of the crystallization enthalpy. **b** X-ray diffraction (XRD) patterns of the as-prepared and heat-treated samples. Both the two samples are fully amorphous within the detection accuracy of XRD. **c** The total scattering factor $S(Q)$ (inset: a close-up of the first sharp diffraction peak) and (**d**) the reduced pair-distribution function $G(r)$ (inset: fitting of the peak height $h(r)$ and peak position $r$ by an exponential decay function $h(r)=A\exp(-r/\xi)$, where $\xi$ is the cutoff length and $A$ refers to the amplitude) of the as-prepared and heat-treated samples obtained by synchrotron XRD at room temperature. **e** High-resolution transmission electron microscopy (HRTEM; scale bar: 5 nm) and selected area electron diffraction (SAED; inset, scale bar: 5 nm⁻¹) results of the as-prepared sample. **f** High-angle annular dark-field (HAADF; scale bar: 10 nm) image of the as-prepared sample. **g**, **h** Same as (**e**, **f**) but for the heat-treated sample. The results confirm that the heat-treated sample remains a single-phase glass. Source data are provided as a Source Data file.

**Table 1 | Properties of the liquid-cooled glass, ultrastable glass, and crystal for TiZrCuNiBe alloys**

|  | Density (g cm⁻³) | Reduced Modulus (GPa) | Hardness (GPa) | $T_{g@1000K/s}$ (K) | $\Delta T_{@10K/min}$ (K) | OR (Å) |
|---|---|---|---|---|---|---|
| **Liquid-cooled glass** | 6.264 ± 0.027 | 111.9 ± 1.1 | 7.64 ± 0.11 | 774 ± 14 | – | 18.3 |
| **Ultrastable glass** | 6.408 ± 0.013 | 142.0 ± 0.9 | 9.63 ± 0.16 | 860 ± 11 | 144 | 25.8 |
| **Crystal** | 6.455 ± 0.009 | 149.7 ± 1.4 | 9.73 ± 0.09 | – | – | – |

$T_{g@1000K/s}$ represents the onset glass transition temperature measured by flash DSC with a heating rate of 1000 K s⁻¹. $\Delta T_{@10K/min}$ denotes the width of the supercooled liquid region measured using DSC at a heating rate of 10 K min⁻¹. OR refers to the ordering-range.

## Densification

Density serves as a crucial indicator for assessing the stability of glasses. The gas displacement method was used to accurately measure the densities of the as-prepared and heat-treated TiZr-CuNiBe ribbons. As shown in Fig. 2a & Table 1, the heat-treated TiZrCuNiBe glass has a 2.3% higher density than that of the as-prepared one, consistent with the shift of $Q_1$ to higher $Q$ in $S(Q)$ and a larger ordering-range analyzed in $G(r)$. Such a huge density

increment induced by GGT was not noticed previously in metallic glasses. As shown in Fig. 2b, the density increment in this work is even slightly larger than that caused by PVD[14,15,21,39–46] and natural aging[10,47] (detailed data is in Supplementary Table 3, with some additional discussion in Supplementary Note 2). This indicates that utilizing the glass-to-glass transition can effectively improve the stability of metallic glasses, which is comparable to that of glasses prepared by natural aging or PVD method.

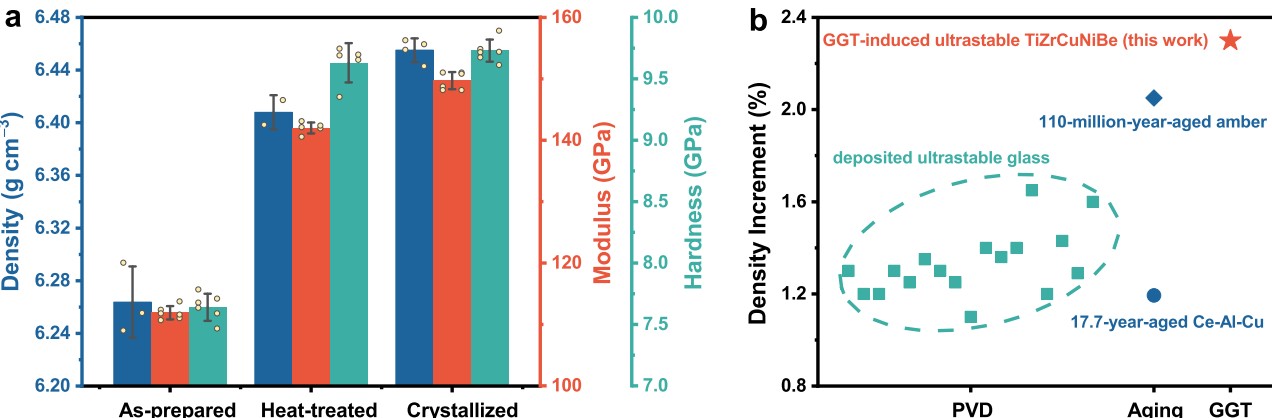

**Fig. 2 | Enhancement of density and mechanical properties in TiZrCuNiBe glassy ribbons via the glass-to-glass transition (GGT). a** Density, reduced modulus, and hardness of the as-prepared, heat-treated, and crystallized samples. GGT leads to a 2.3% increase in density, a 26.9% increase in modulus, and a 26.0% increase in hardness. Error bars represent standard deviations from parallel measurements. Source data are provided as a Source Data file. **b** Comparison of density increments resulting from physical vapor deposition (PVD)[14,15,21,39–46], natural aging[10,47], and GGT. Detailed data can be found in Supplementary Table 3.

In the literature, a parameter $\theta_K$ ($\theta_K = (T_g - T_f)/(T_g - T_K)$, where $T_g$, $T_f$, and $T_K$ are referred to as the glass transition temperature, fictive temperature and Kauzmann temperature, respectively) was often used to measure how far away an ultrastable glass is from the ideal glass[11]. The parameter is based on the estimation of $T_f$, which involves extrapolating the properties of the supercooled liquid to below $T_g$. In this study, however, due to the exothermic event in the supercooled liquid, another supercooled liquid with distinct thermodynamic state could form. This makes the extrapolation method inapplicable, thus making it difficult to determine $T_f$ and further $\theta_K$. Considering this, based on density measurement, we proposed a new parameter $\theta_\rho$ as an alternative:

$$\theta_\rho = \frac{\rho - \rho_{LCG}}{\rho_{crys} - \rho_{LCG}} \qquad (2)$$

where $\rho$, $\rho_{LCG}$, and $\rho_{crys}$ represent the densities of the glass to be evaluated, the liquid-cooled glass, and the corresponding crystal, respectively. For the same composition, the density of the ideal glass should be the highest among all glassy states, but it is not supposed to exceed that of its stable crystalline counterpart, which sets an upper boundary of the densities of all glassy states. The liquid-cooled glass, on the other hand, has a density close to the lowest, which could be regarded as the lower boundary of the densities. For a glass, $\theta_\rho$ ranges between 0 and 1, where a larger value indicates a state closer to the ideal glass state. For the heat-treated TiZrCuNiBe glass, $\theta_\rho$ equals 0.75, indicating that it has proceeded 75% toward the ideal glass state.

A much denser amorphous packing results in a much higher modulus and hardness. The reduced modulus and the hardness are measured by nanoindentation tests and the typical load-displacement curves are shown in Supplementary Fig. 4. The modulus of the heat-treated glass is 26.9% higher than that of the as-prepared one, and only 5.1% lower compared to its crystalline counterpart (Fig. 2a & Table 1). It illustrates that the ultrastable TiZrCuNiBe glass can overcome the modulus softening phenomenon to a large extent. Additionally, the hardness of the heat-treated glass increased by 26.0% compared to the as-prepared one.

### Thermal stability

The heat-treated TiZrCuNiBe glass has high thermodynamic and kinetic stability. Thermodynamically, the heat-treated sample exhibits much lower enthalpy, attributed to the enormous exothermic event. Typically, the enthalpy reduction from a liquid-cooled glassy state to

an ultrastable glassy state is about $5 - 10\,J\,g^{-1}$[7]. In this work, the enthalpy reduction measures as large as $15.5\,J\,g^{-1}$ (accounts for 45.6% of its total crystallization exotherm), confirming that the newly-formed glass is situated in the domain of ultrastable glasses. The greatly decreased enthalpy induced by GGT designates the significantly enhanced thermodynamic stability.

To determine the onset glass transition temperature of the as-prepared sample, flash DSC analyses were carried out with a heating rate of $1000\,K\,s^{-1}$, of which the increased sensitivity will highlight the glass transition. Both the as-prepared and heat-treated samples exhibit obvious glass transition, while the $T_g$ of the heat-treated sample is 11.1% higher than that of the as-prepared one (see Table 1, typical flash DSC curves and the detailed description can be seen in Supplementary Fig. 5 and Supplementary Note 1, respectively), indicating that a much higher temperature is required to dislodge the atoms from the trapped glassy configuration into a mobile equilibrium supercooled liquid. In addition, the supercooled liquid region of the heat-treated TiZrCuNiBe glass is as large as 144 K (see Table 1, obtained from the DSC curve at a heating rate of $10\,K\,min^{-1}$), surpassing that of any other reported ultrastable MGs[16,18,27–30,47–49], indicating a stable supercooled liquid with great difficulty to be transformed into crystals (note that the glass may be suitable for thermoplastic processing or nanoimprinting).

The increment of the onset glass transition temperature ($\delta T_g = (T_{g,L} - T_{g,H})/T_{g,H}$, subscripts 'L' and 'H' mean low- and high-energy state, respectively) and the width of the supercooled liquid region ($\Delta T$) of the GGT-induced[27,28,30,49], vapor-deposited[16–18,48], and naturally aged[47] metallic glasses are summarized in Fig. 3 (detailed data can be seen in Supplementary Table 5). Both $\delta T_g$ and $\Delta T$ of the GGT-induced low-energy MGs are comparable to those of vapor-deposited and naturally aged ultrastable MGs, confirming GGT as an effective way to improve the kinetic stability. Among these highly stable MGs, the GGT-induced TiZrCuNiBe glass exhibits high $\delta T_g$ and $\Delta T$ (located at the top right corner of the diagram), underscoring its superior kinetic stability.

To further verify its crystallization resistance and mechanical stability, the heat-treated TiZrCuNiBe ribbon was annealed at $T_g + 70$ K for 10 h. There appear no sharp Bragg peaks (Supplementary Fig. 6a) and no crystallization enthalpy decline (Supplementary Fig. 6b) after long-time annealing, designating no crystallization and high crystallization resistance. Additionally, only minimal changes were observed in hardness and modulus (Supplementary Fig. 6c), manifesting good mechanical stability for the heat-treated TiZrCuNiBe glass. The results above further confirm the high kinetic stability of the newly-formed glass induced by GGT.

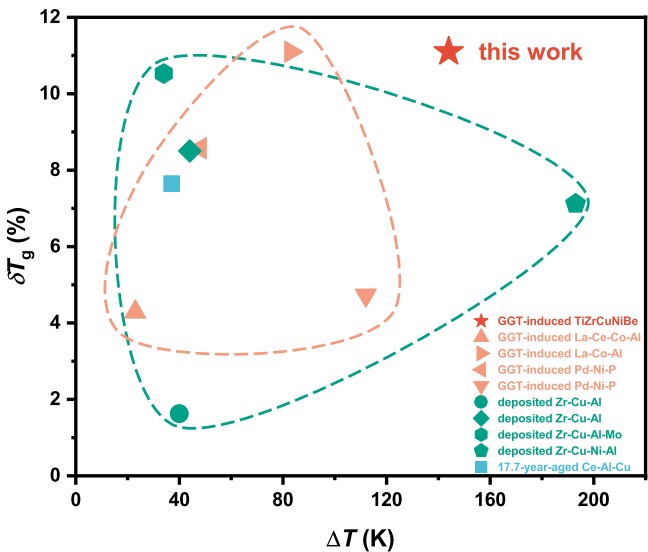

**Fig. 3 | Kinetic stability of the ultrastable TiZrCuNiBe glass induced by the glass-to-glass transition (GGT).** The increment of the onset glass transition temperature, $\delta T_g$, is defined as $\delta T_g = (T_{g,L} - T_{g,H})/T_{g,H}$, where $T_{g,L}$ and $T_{g,H}$ denote the onset glass transition temperature of the low- and high-energy glass, respectively. $\Delta T$ represents the width of the supercooled liquid region for the low-energy glass, determined from differential scanning calorimetry (DSC) tests at a heating rate of 10/20 K min[−1]. The kinetic stability of the heat-treated TiZrCuNiBe glass is compared with that of other low-energy metallic glasses, including those induced by GGT[27,28,30,49] (enclosed within the orange dashed line), prepared via physical vapor deposition[16–18,48] (enclosed within the green dashed line), and produced through natural aging[47]. The ultrastable TiZrCuNiBe metallic glass induced by GGT exhibits a significantly enhanced $T_g$ and a large $\Delta T$, indicating its superior kinetic stability. Detailed data can be found in Supplementary Table 5.

## Discussion

The $G(r)$ analyses reveal that the glassy phase induced by GGT can be reserved to RT with a more ordered structure than that of the liquid-cooled one. Similar phenomena have also been observed in some other metallic glasses[26,28,29,32], but most of them concentrated on the local structural changes caused by the phase transition rather than the nature of the newly-formed glasses. Here, we argue that an ultrastable glass could be obtained through the GGT in TiZrCuNiBe metallic glass.

One of the most important indicators of obtaining ultrastable glasses is the apparent increased density. For vapor-deposited ultrastable glasses, the density increment was typically 1 - 2% by evaluating the thickness change of the film (Fig. 2b & Supplementary Table 3). Natural aging could also achieve a similar densification effect (Fig. 2b & Supplementary Table 3), but it requires tens to millions of years[10,47]. Previous studies on GGT, however, either overlooked density changes[29,30,32], or failed to observe significant densification, which is likely due to the inapparent structural changes induced by the weak heat release[27,28,50]. In this study, the huge heat release (Fig. 1a), apparent structural changes (Fig. 1c, d), and extended ordering-range noticed in the GGT of TiZrCuNiBe glass, demonstrated a 2.3% higher density than that of the liquid-cooled one. The great density improvement suggests enhanced stability. It is noteworthy that such a significant increase in density is achieved through a simple annealing treatment, which is much faster than the natural aging process, and is also different from the PVD method which can only produce thin films on a two-dimensional scale. Unsurprisingly, the higher density also correlates with the significantly increased modulus and hardness, affirming the formation of stronger atomic bonds via GGT-induced densification. Additionally, the kinetic stability of the GGT-induced TiZrCuNiBe glass was revealed by the elevated $T_g$, a broad supercooled liquid region (Fig. 3 & Table 1), and the excellent resistance to crystallization

(Supplementary Fig. 6). All the results confirm the ultrastability of the TiZrCuNiBe glass induced by GGT.

Approaching the ideal glass state has always been the relentless pursuit of physicists on amorphous materials, which is of great significance in revealing the nature of glass transition[2]. Then, to measure how far away the presented ultrastable glass is from the ideal glass, we propose a figure of merit $\theta_\rho$ based on the density difference (Eq. (2)). The closer the value of $\theta_\rho$ approaches 1, the closer the glass state approaches the ideal glass state. Then, the GGT-induced ultrastable TiZrCuNiBe metallic glass, with a $\theta_\rho$ of 0.75, proceeds 75% toward the ideal glass state. For vapor-deposited and naturally aged glasses, $\theta_K$ was used in the literature to quantify the proximity to the ideal glass state. Fortunately, both $\theta_\rho$ and $\theta_K$ serve as indicators of the proximity between a glass state and the ideal glass state, though their definitions are based on different physical quantities. It is noticed that the values of $\theta_K$ are 0.61 and 0.71 for the 20-million-year-old amber[9] and the 17.7-year-aged Ce-Al-Cu metallic glass[47], respectively; for the PVD method, the typical value of $\theta_K$ is around 0.50[4,11,51,52]. Compared to ultrastable glasses obtained by PVD and natural aging, the ultrastable TiZrCuNiBe glass demonstrates a comparable or even better effect in terms of approaching its ideal glass state.

Additionally, only a few metallic glasses up to now were reported to exhibit a GGT, which raises a question of whether the GGT is a general method, like natural aging and PVD, to access ultrastable glass state. To solve this concern, we designed the composition dependence experiments, and (TiZrCuNi)$_{100-x}$Be$_x$ (x = 8, 12, 16, 20, and 24, labeled as B8, B12, B16, B20, and B24, respectively) glassy ribbons were prepared and investigated. Figure 4a shows the DSC curves of the as-prepared samples, and the corresponding heat-treated samples were obtained by heating the as-prepared samples to $T_{HT}$ and then cooling down to RT, whose XRD patterns are shown in Fig. 4b. For B16 and B24 glasses, GGT persists, and the heat-treated samples are fully amorphous (HRTEM and SAED images seen in Supplementary Fig. 7). Besides, it is noticed that the second exothermic peak shifts to higher temperatures as the Be content increases from 16 to 24 at.%, so that a platform as wide as 40 K appears between the first and second peaks of B24 glass. By contrast, when the Be content drops below 16 at.%, some other noticeable and intriguing changes emerge: the second exothermic peak diminishes while the first exothermic peak intensifies with decreasing Be content, indicating that a portion of the exothermic heat originally associated with the second peak merging into the first peak. Accordingly, minor crystallization occurs in the heat-treated samples for B12 and B8 glasses (Fig. 4b), further confirming the overlap of the GGT and crystallization. The composition dependence experiments reveal a competitive relationship between GGT and crystallization, which determines the phase transition sequence.

The results of the composition dependence of the GGT can be explained from the perspective of the driving forces at the onset of the phase transitions. Figure 4c&d schematically illustrate the Gibbs free energy curves of a high-energy glass (G1, the liquid-cooled glass), low-energy glass (G2, a glass induced by GGT), and crystalline phase (X), respectively. The high-energy glass G1 may undergo two potential transformations, namely, a glass-to-glass transition to form a low-energy glass G2 or the precipitation of the crystalline phase X. The sequence of the two processes is determined by the driving forces at the onset of the phase transitions. The driving force is defined as $-dG/dn$, where $dG$ is the change in the Gibbs free energy when the amount of substance of the new phase increases by $dn$ at the onset of a phase transition. As shown in Fig. 4c, d, the driving forces of GGT and crystallization can be represented by different arrows, with the length indicating the magnitude of the driving force, and the direction indicating whether the driving force is positive or negative (downward denoting positive driving force, and vice versa).

For a liquid-cooled glass with composition $c_1$ (Fig. 4c), the driving force of the precipitation of X from G1 is negative while that of the GGT

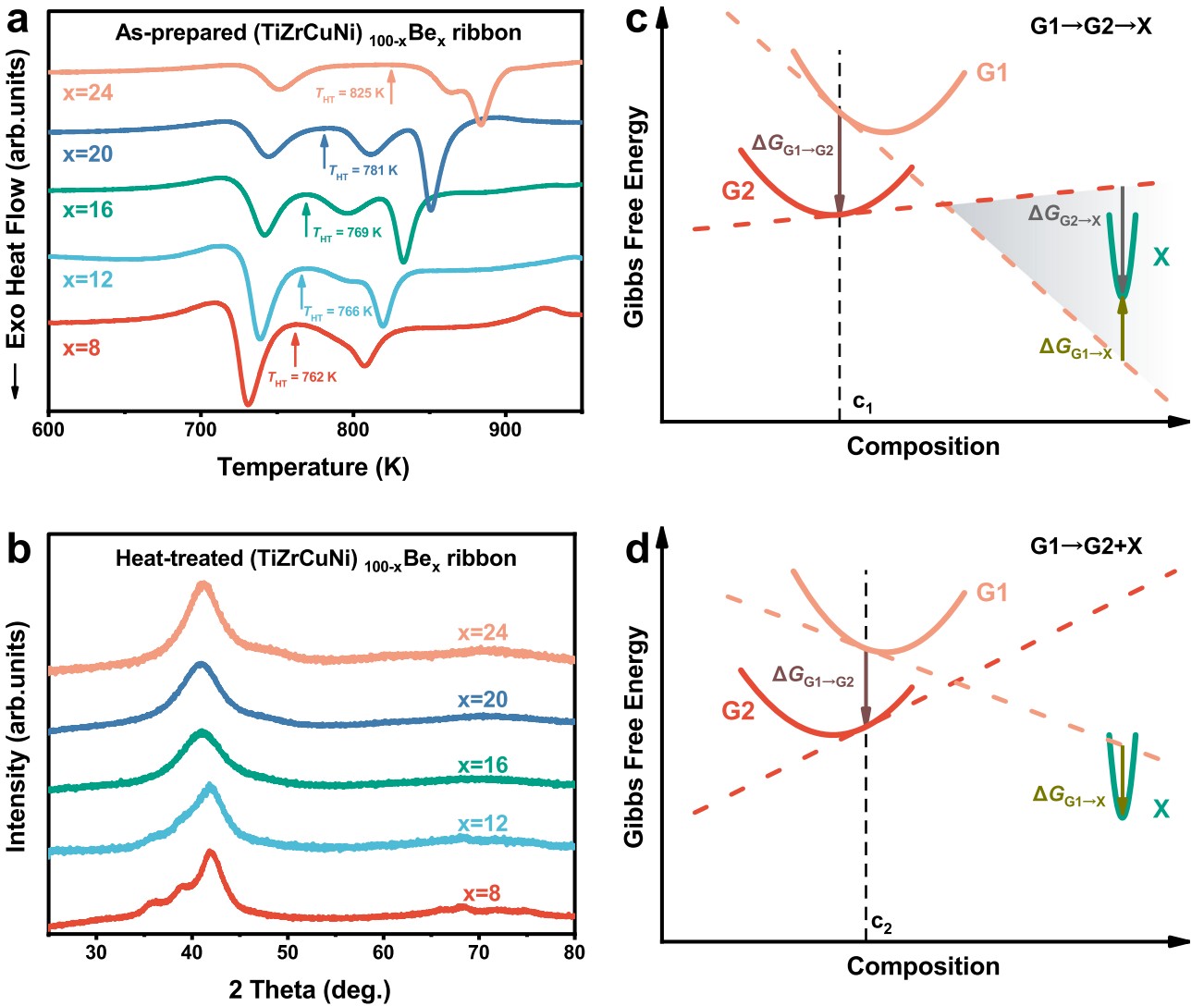

**Fig. 4 | Composition dependence of the glass-to-glass transition (GGT).**
**a** Differential scanning calorimetry (DSC) curves of the as-prepared
$(TiZrCuNi)_{100-x}Be_x$ (x = 8, 12, 16, 20, and 24) ribbons with a heating rate of
10 K min$^{-1}$. **b** X-ray diffraction (XRD) patterns of the heat-treated $(TiZrCuNi)_{100-x}Be_x$
(x = 8, 12, 16, 20 and 24) ribbons. For each composition, the heat-treated sample
was obtained by heating the as-prepared sample to $T_{HT}$ and then cooling it down to
room temperature. **c**, **d** schematically illustrate the composition dependence of
GGT from the perspective of the driving forces at the onset of the phase transitions.
G1, G2 and X represent the high-energy glass, low-energy glass, and crystalline
phase, respectively. Their Gibbs free energy curves are indicated by solid lines in
orange, red, and green, respectively, while the tangent lines to the free energy

curves at a specific composition are shown as dashed lines in orange, red, and
green, respectively. The driving forces of different transitions can be represented
by different arrows, with the length indicating the magnitude of the driving force,
and the direction indicating whether the driving force is positive or negative
(downward denoting positive driving force, and vice versa). **c** The driving force of
the precipitation of X from G1 is negative while that of GGT is positive, therefore the
GGT occurs before crystallization. That is, a well-separated GGT can be observed
when the minimum of the Gibbs free energy for X phase falls within the shaded
area. **d** The driving force of the precipitation of X from G1 becomes positive,
resulting in the overlap of the GGT and crystallization. Source data are provided as a
Source Data file.

is positive, resulting in a well-separated GGT before crystallization.
Afterwards, the precipitation of X from G2 exhibits a positive driving
force, resulting in a subsequent crystallization. In other words, only
when the minimum of the Gibbs free energy of X lies above the tangent
line at $c_1$ of G1 but below the tangent line at $c_1$ of G2 (the shaded area in
Fig. 4c), can the well-separated GGT be observed, which are the cases
for the B16, B20 and B24 glasses. Further on, the influence of Be con-
tent for the three glasses is illustrated in Supplementary Fig. 8a. As the
Be content increases (the composition point shifts to the left), the
driving force of the precipitation of X from G2 decreases, thereby
causing a suppressed crystallization process (the onset temperatures
of the second exothermic peaks increase).

When the composition changes to $c_2$ (Fig. 4d, which are the cases
for B12 and B8 glasses), in addition to GGT, the precipitation of X from

G1 also gains a positive driving force. In such a situation, the GGT and
crystallization could occur at the same time, which is the reason why
the heat-treated samples slightly crystallize and the first exothermic
peak intensifies in the DSC curve. Afterwards, a few crystals are further
precipitated from G2, corresponding to the remaining second exo-
thermic peak (Fig. 4a). When the composition changes further, it can
be expected that the driving force of the precipitation of X from G1 will
be much larger than that of GGT (as seen in Supplementary Fig. 8b). In
this case, the crystallization will dominate over GGT so that no GGT
could be observed, which is probably for most cases.

The above results and analyses on the composition dependence
of GGT suggest the reason why GGT is usually invisible. In most cases,
GGT cannot be observed because its driving force is much smaller than
that of crystallization. Only when a positive driving force of GGT and a

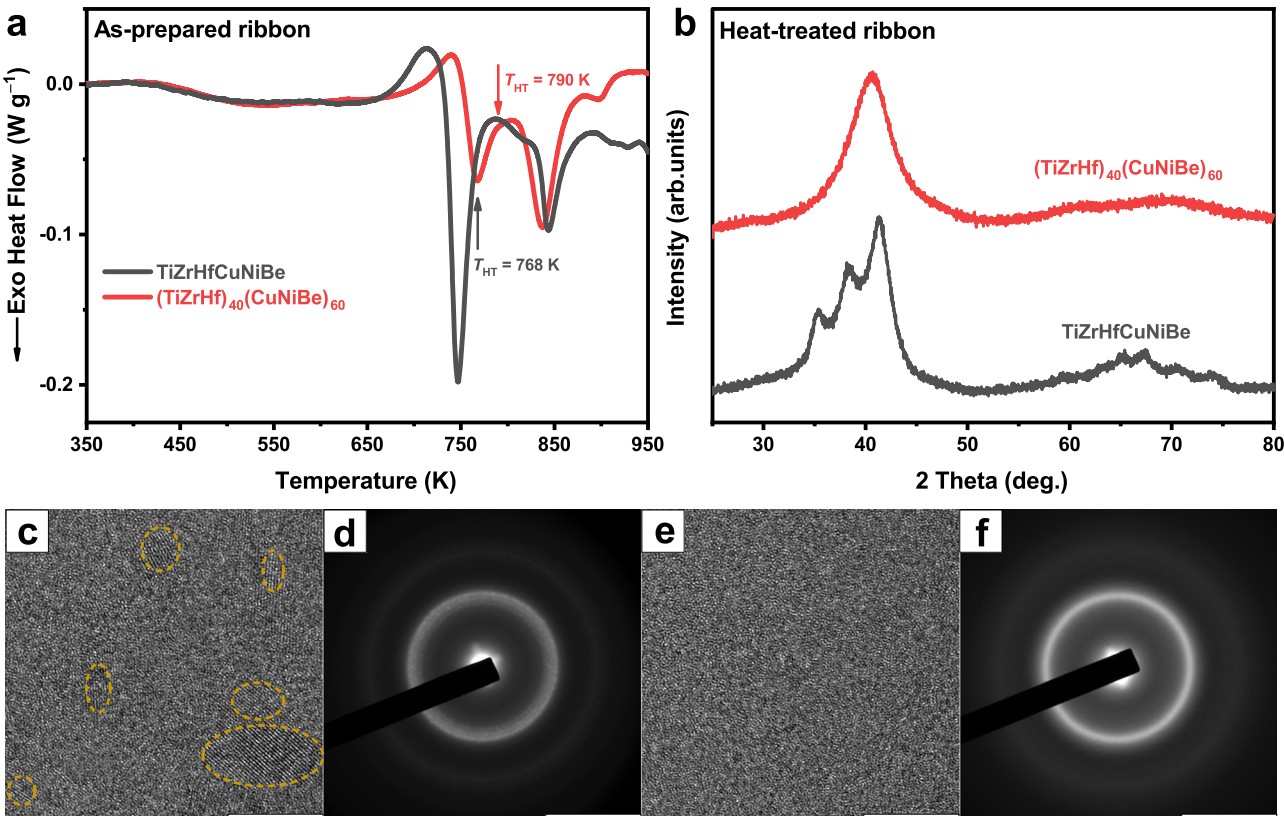

**Fig. 5 | Observation of the well-separated glass-to-glass transition in Ti-Zr-Hf-Cu-Ni-Be metallic glass by composition adjustment. a** Differential scanning calorimetry (DSC) curves of the as-prepared TiZrHfCuNiBe and $(TiZrHf)_{40}(CuNiBe)_{60}$ ribbons with a heating rate of 10 K min$^{-1}$. The heat-treated samples were prepared by heating the as-prepared samples to $T_{HT}$ and immediately cooling down. **b** X-ray diffraction (XRD) patterns of the heat-treated TiZrHfCuNiBe and $(TiZrHf)_{40}(CuNiBe)_{60}$ ribbons. The former is partially crystallized while the latter is fully amorphous. **c** High-resolution transmission electron microscopy (HRTEM; scale bar: 5 nm) and (**d**) selected area electron diffraction (SAED; scale bar: 5 nm$^{-1}$) images of the heat-treated TiZrHfCuNiBe ribbons. Nano-crystals (marked by yellow dashed circles) were observed dispersed in the amorphous matrix, corresponding with the XRD spectrum. **e, f** Same as (**c, d**) but for the heat-treated $(TiZrHf)_{40}(CuNiBe)_{60}$ ribbons. No nano-crystals were observed, confirming the fully amorphous structure. Source data are provided as a Source Data file.

negative driving force of crystallization are satisfied concurrently (as shown in Fig. 4c), can a well-separated GGT be observed. This indicates that GGT could be a universal phenomenon but is often covered up by the strong crystallization process. The key to achieving a well-separated GGT lies in adjusting the competitive relationship between GGT and crystallization by various means. Consequently, the different arguments in the literature regarding the circumstances under which GGT will emerge could be unified. For instance, introducing non-metallic elements to increase the proportion of covalent bonds[27], or utilizing the high-entropy effect[29,32], actually serves as different means to suppress crystallization and promote GGT.

Adjusting the composition could also impact the relative competitive relationship between GGT and crystallization as illustrated in Fig. 4c, d, which suggests a convenient way to explore more metallic glasses with well-separated GGT. If only a small number of (nano-)crystals formed after a large exothermic event (broad amorphous peaks superimposed with some small Bragg diffraction peaks in the XRD spectrum), the alloy is likely to experience the GGT and the crystallization at the same time. In such cases, it is highly possible to obtain a well-separated GGT via composition adjustments. For example, through literature review, we noticed an equal-atomic TiZrHfCu-NiBe glass[53] and then prepared and investigated it. The heat-treated sample was obtained by heating the as-prepared sample to $T_{HT}$ (the end temperature of the first post-$T_g$ exothermic peak) and then cooling down to RT. Partial crystallization was observed in the XRD spectrum (Fig. 5b), HRTEM (Fig. 5c), and SAED (Fig. 5d) images of the heat-treated sample, similar to the case of the heat-treated $(TiZrCuNi)_{92}Be_8$

sample. Instead of attributing the first post-$T_g$ exothermic peak to crystallization entirely, we suggest that the GGT and crystallization may occur at the same time, causing the exothermic event. In such a case, the driving force of the GGT and crystallization are both positive as Fig. 4d illustrated, thereby leading to no observation of a well-separated GGT. Then adjusting the alloy composition from equal-atomic to $(TiZrHf)_{40}(CuNiBe)_{60}$, we successfully reveal a well-separated GGT, with no crystallization or nano-crystals precipitation observed in XRD spectrum (Fig. 5b), HRTEM (Fig. 5e) and SAED (Fig. 5f) images of the heat-treated sample. The case of Ti-Zr-Hf-Cu-Ni-Be alloy system here serves as a reminder for us to reexamine the crystallization behavior of metallic glasses reported in the literature, given the potential hints of GGT. Also, this case again supports our statements, and we believe that adjusting the composition would be an effective method to discover more well-separated GGT and therefore the ultrastable metallic glasses.

Thus, an easy, effective, and general way to explore and access ultrastable glasses has been demonstrated. Owing to simple atomic packing with non-directional metallic bonds, metallic glasses may be more adept than net-working glasses in overcoming the kinetic barriers hindering the transition to a low-energy state. For instance, the formation of a lower-energy glass phase (G-phase) was observed when cooling simple Ag, Cu-Ag, and Cu-Zr atomic liquids using molecular dynamics simulations[6,54-56]. From the perspective of the potential energy landscape (Fig. 6), through the glass-to-glass transition, the substantial barrier between two megabasins was surmounted and the glass transits from a high energy state to a much lower one, thereby

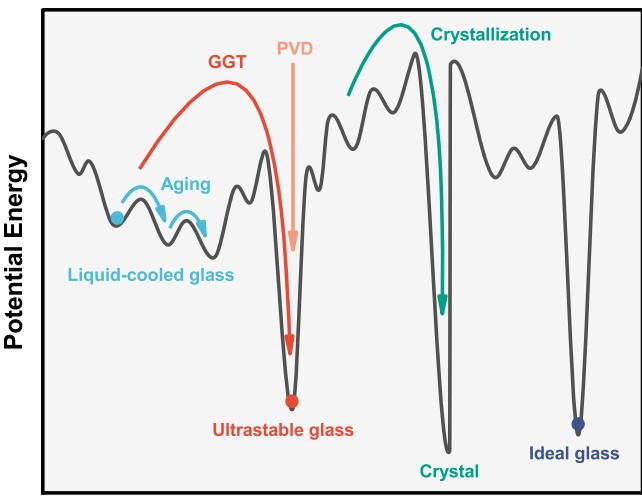

**Fig. 6 | Schematic illustration of some transitions between different glassy states on a potential energy landscape.** The x-axis represents all configurational coordinates, with GGT and PVD denoting glass-to-glass transition and physical vapor deposition, respectively.

approaching the ideal glass. By contrast, it is of great difficulty for aging to surmount such a high barrier between two megabasins unless an exceptionally long period of time is involved. Both the PVD and the GGT method successfully circumvent this long timescale, but through different pathways. The former leverages rapid surface diffusion, allowing molecules/atoms to quickly rearrange into lower-energy configurations, without altering the supercooled liquid. The latter utilizes a phase transformation to produce a lower-energy supercooled liquid, which is then frozen to RT to form a more stable glass. Compared with natural aging and PVD, employing GGT for preparing ultrastable glasses offers comparable stability and demonstrates a comparable ability to approach the ideal glass state. Furthermore, GGT has additional advantages: (1) as a thermal effect, GGT only requires a simple annealing process within a few hours to achieve effects that natural aging takes tens of hundreds of years to accomplish; (2) as a post-processing method, GGT requires no substrate and imposes no size limitations; (3) as a bulk process, GGT induces uniform structural changes without anisotropy.

Obtaining ultrastable glasses through natural aging has been a formidable challenge due to its inherently sluggish dynamics. The PVD technique has made significant strides by creating two-dimensional ultrastable thin film glasses, leveraging rapid surface diffusion to reduce the preparation times to a few tens of hours. Building upon these advancements, we introduce a general pathway to obtain ultrastable glasses within a couple of hours through a bulk transformation process known as the glass-to-glass transition. Based on this, more ultrastable glasses are expected to be discovered, helping resolve the instability issues of glass materials and devices, and acquire more knowledge about the nature of glass transition.

## Methods
### Sample preparation
The ingots were prepared by arc melting the mixtures of pure elements (purity ≥ 99.9 wt.%) under a Ti-gettered high-purity argon atmosphere. The ingots were remelted at least five times to ensure chemical homogeneity. Ribbons with a thickness of ~35 μm were prepared by melt-spinning method with a single-copper roller at a wheel surface velocity of 40 m s⁻¹ under a high-purity argon atmosphere. The heat-treated samples were produced in the differential scanning calorimeter by heating the as-prepared samples with a heating rate of

10 K min⁻¹ to $T_{HT}$ (the end temperature of the first post-$T_g$ exothermic peak), and then immediately cooling down to room temperature.

### Differential scanning calorimetry (DSC)
DSC curves were measured using a synchronous thermal analyzer (STA449 F3, NETZSCH, Germany) with a heating rate of 10 K min⁻¹ under a high-purity argon atmosphere. Each sample weighed ~20 mg and was packed tightly in an $Al_2O_3$ crucible. For each sample, the DSC curve of the fully crystallized sample (sample heated to 973 K) was used as a baseline.

### X-ray diffraction (XRD)
XRD tests were conducted using an X-ray diffractometer (D/max-2550, Rigaku, Japan) with Cu Kα radiation at RT. The $\theta - 2\theta$ mode was adopted with a scanning rate of 2 degrees min⁻¹.

### Transmission electron microscopy (TEM) analyses
TEM samples were thinned by an ion mill (PIPS model 691, Gatan, U.S.) at 77 K. The ion energy and incidence angle were set to 5 keV and 5°, respectively, until a hole was formed. Final polishing was conducted at a reduced voltage and incidence angle of 2 kV and 3°, respectively. The selected area electron diffraction (SAED), high-resolution TEM (HRTEM), and high-angle annular dark-field (HAADF) analyses were performed by a high-resolution transmission electron microscopy (JEM-2100F, JEOL, Japan) at 200 kV. HAADF images were acquired under scanning transmission electron microscopy (STEM) mode with a probe size of ~ 1.5 nm.

### Synchrotron X-ray diffraction and pair-distribution functions analyses
Synchrotron X-ray diffraction experiments of the as-prepared and heat-treated TiZrCuNiBe glassy ribbons were conducted at the beamline I15-1 of the Diamond Light Source, UK. The wavelength of the X-ray beam is 0.161669 Å, with a beam size of 490 μm × 10 μm. The one-dimensional diffraction patterns were obtained by integrating the 2D diffraction images using Fit2D software[57]. The total scattering factor $S(Q)$ and the reduced pair-distribution function $G(r)$ were obtained using PDFgetX3 software package[58]. Considering the asymmetry of the peak shape, the position of the FSDP was determined by

$$Q_1 = \frac{\int Q \cdot S(Q)dQ}{\int S(Q)dQ} \tag{3}$$

The integral width of the FSDP was determined by

$$W = \frac{\int S(Q)dQ}{S_p} \tag{4}$$

where $S_p$ is referred to the peak height of the FSDP.

### Atom probe tomography (APT) analyses
Needle-shaped specimens required for APT were fabricated using electropolishing method at 5 − 15 V. 20% perchloric acid ethanol solution and 5% perchloric acid butyl cellosolve solution were used for initial and final polishing, respectively. The 3D atom probe tomography (APT) characterization was performed in a CAMECA Instruments LEAP 4000X HR local electrode atom probe. The specimens were analyzed at 50 K using laser mode with a laser energy of 60 pJ, a pulse rate of 200 kHz, and an evaporation detection rate of 1%. Imago Visualization and Analysis Software (IVAS) version 3.6.8 was used for creating the 3D reconstructions and data analyses.

### Density measurement
The densities were measured using the gas displacement method (99.999% helium was used as the displacement medium) with a TD-

2200 gas pycnometer (BUILDER, China) at $35 \pm 0.01\,°C$. The mass of each sample used for the measurement is no less than 1.5 g. Three parallel samples were tested. For each test, the skeletal volume was tested four times.

## Nanoindentation analyses

Nanoindentation analyses on the as-prepared, heat-treated, and crystallized samples were performed using an Agilent Nano Indenter G200 with a diamond Berkovich indenter at RT. 6 indents were made for each sample to ensure repeatability. During each indentation test, the load was imposed with a constant loading rate of $1\,mN\,s^{-1}$ until the maximum load of 30 mN was reached. The maximum load was held for 5 s before unloading with $1\,mN\,s^{-1}$.

## Flash DSC analyses

A Mettler-Toledo Flash DSC 2+ was used to analyze the thermal behaviors of the as-prepared and heat-treated TiZrCuNiBe ribbons under flash heating. Each sample with a size of about $50\,\mu m \times 50\,\mu m$ was cut from the corresponding ribbon and then loaded on a UFH 1 chip sensor. Under the protection of pure argon at a flow rate of $80\,ml\,min^{-1}$, each sample was heated to 800 °C with a heating rate of $1000\,K\,s^{-1}$. At least three parallel samples were measured to ensure repeatability.

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

## Acknowledgements

The authors would like to thank Prof. Jingfeng Li in Tsinghua University for the help in the DSC tests. We also thank Dr. Hongjie Xu and Xinglong Yang for helpful discussions. This work was supported by National Natural Science Foundation of China (Grant Nos.: 52471176, 52271148) and National Key Basic Research and Development Program of China (Grant Nos.: 2022YFB3804100, 2022YFB4200800). Q.Z. acknowledges the support from Shanghai Key Laboratory of Material Frontiers Research in Extreme Environments, China (No. 22dz2260800), the Shanghai Science and Technology Committee, China (No. 22JC1410300).

## Author contributions

H.B., J.J., Y.W., and L.S. conducted the sample preparation. H.B. conducted DSC, XRD and flash DSC experiments and analyses. H.B., J.K., and Y.S. conducted TEM experiments and analyses. Q.Z. and H.B. conducted synchrotron XRD experiments and analyses. H.B. and W.G. conducted APT analyses. H.B. and H.L. conducted density measurements. H.B. and Y.S.S. conducted nanoindentation measurements. All authors participated in the discussion and interpretation of the results. H.B. and Y.S. wrote and revised the manuscript. P.G., H.L., J.K., H.D., H.-S.C., R.W., and K.Y. commented on the manuscript. Y.S. and K.Y. supervised the research project and coordinated the research collaborations.

## Competing interests

The authors declare no competing interests.
