## [Transparent Peer Review file · Nature Communications]

Accessing ultrastable glass via a bulk transformation

Corresponding Author: Professor Yang Shao

Version 0:

Reviewer comments:

Reviewer #1

(Remarks to the Author)

This is a fascinating study of glass formation in a TiZrCuNiBe metallic alloy. The authors show that this system has a glass-to-glass transition (or liquid-to-liquid transition). They heat-treat the as-prepared glass to induce this transition, and then cool back to the glassy state. As a result, they prepared a glass that is high density and low enthalpy, relative to the initial rapidly-cooled glass. Interestingly, when the new glass is heated it yields a supercooled liquid that is quite stable against crystallization.

The authors make extensive comparison to ultrastable glasses prepared by physical vapor deposition (PVD). By some metrics, the bulk processed glasses prepared here are more stable than PVD glasses. This provides a very interesting point of comparison for this new work.

This manuscript is very clearly written and the experimental work appears to be of very high quality. I am excited to read about these new results and I am inclined to support publication in Nature Communications. I do have some important concerns, and I hope to be able to give my endorsement for publication after seeing how the authors respond to these concerns.

Major issues:

The comparison to PVD glasses is definitely relevant and important, but I think it requires a bit more explanation. For PVD glasses made from organic molecules, there is only one supercooled liquid – all the properties of the PVD glass are measured relative to a liquid-cooled glass of the same supercooled liquid. In the present manuscript, the heat-treated glass is a liquid-cooled glass, and its properties are compared to a liquid-cooled glass of a different supercooled liquid. From this perspective, the new glass that is prepared in this work is not so remarkable. Rather, it is the second supercooled liquid that is remarkable.

In the abstract and elsewhere, the authors propose that their approach is universal for metallic glasses. Yet they show several compositions where it does not work (without adjusting the composition), therefore it is not universal for any given composition. I suggest that the authors drop the claim of universality and describe this as a “general” approach.

Minor issues:

In the abstract, the anisotropy of PVD glasses is described as a limitation. For some applications (organic electronics), anisotropy improves performance – so it does not seem fair to call this a limitation. It is a difference. Isotropy may be preferred in some applications.

About this sentence: “Ideal glass is expected to have a density approaching infinitely that of its crystalline counterpart...” I think you are completely incorrect on this point.

For organic liquids, typical extrapolation of supercooled liquid properties show that the configurational entropy goes to zero (defining the ideal glass) at a temperature where the supercooled liquid is less dense than the crystal. If you are making a statement about metallic glasses in particular, please provide some references that support your claim.

If I understand you correctly, the heat-treated glass is about 0.7% less dense than the crystallized sample. Please compare this number to other literature reports for metallic glasses. Is 0.7% unusually small compared to other metallic glasses? Is 3% (the difference between the ribbon sample and crystal) unusually large? Please include a careful comparison with the literature.

Figure 2b seems not quite fair. For the PVD glasses, the density increase is calculated relative to a glass made by cooling the liquid at 1 K/min. For your sample, the density increase is calculated relative to the ribbon cooled sample (10^4 K/s??). The difference in cooling rates unfairly makes your result relatively larger. This likely accounts for only a part of the difference, but should be noted.

The comparison of θ_K and θ_ρ cannot be quantitatively meaningful. See first major issue above.
Mark Ediger

Reviewer #2

(Remarks to the Author)

Ultrastable glasses through bulk phase transformation
Ke Fu Yao

This paper reports on using a glass-to-glass transition to fabricate ultrastable glasses.

The research is exciting as it suggests a pathway to reach those "ultraold" glasses through a different mechanism, also different from the recently explored sputtering approach where surface mobility has been used to get into such deep enthalpic states.

Glass to glass transitions have been reported in recent years (a lot of work has come out of Bill Johnson's group at Caltech which the authors should also acknowledge) but has not been associated with ultrastable glasses.

This is the main point of this paper. Clearly the authors provide a multitude of evidence for this claim and I would argue that all there evidence clearly point towards a highly in enthalpy reduced glass.

However, and this is the main comment I have is that if indeed the glass is an ultrastable glass that would mean it would continuously relate to a typical, less stable glass, through time scales of relaxation. In other words, how do we know that it is "the same glass" and not a different phase.

A different glass could be something like a different crystalline material (BCC vs FCC) and the same would be a supersaturated BCC which would reduce in saturation.

Fundamentally, and I strongly believe this work has potentially deep fundamental ramification (and not technological), such difference is important.

In other words, is the low enthalpy glass separated by a nucleation process or is it continuous?

To test such continuous behaviour the author should anneal their glasses at different temperatures for longer than the relaxation time. The relaxation time can be estimated from similar glasses (~ 100 sec at T_g and shortening \sim by order of magnitude every 20 K increase).

This would allow some continuous sampling of properties such as density or moduls. I strongly encourage the authors to carry out such experiments as the outcome reveals if the glass is continuous connected or separated through a nucleation process.

In addition to this some minor comments:

-the authors should cite Bill Johnson work on glass to glass transition

-I found the mentioning of technological relevance of this work far stretched and unnecessary. To me, this is pure science and potentially very exciting as it challenges our fundamental understanding of relaxation phenomena in glasses. Not every work has to be technologically relavant and by mentioning both, the scientific importance is diminished.

-Further, ultrastable glasses are considered to have a very low fictive temperature. Fictive temperature has been directly correlated with mechanical properties (see [J. Ketkaew, Mechanical glass transition revealed by the fracture toughness of metallic glasses, Nat Commun 9 (2018).] [S.V. Ketov, Rejuvenation of metallic glasses by non-affine thermal strain, Nature 524(7564) (2015) 200-+.

] [S. Sohn, A framework for plasticity in metallic glasses, Materialia 31 (2023) 101876.] and the author should mention this correlation (structure change through T_f results in mechanical property change and acknowledge this work.

-the fact that you can fabricate the ultrastable glass "free from substrate" is trivial compare to The fact that it happens in bulk so quickly which is a very big finding.

-line81: "proceeding 75% toward the ideal glass state." How do you know the 100% ideal ideal glass?

Reviewer #3

(Remarks to the Author)

Although the PVD method has been used to prepare amorphous thin films for a long time, it was not until 2007 that Ediger successfully generated ultrastable glasses just by elevating the substrate temperatures.

Similarly, this work is not the first report of the thermally induced polyamorphous transition (glass-to-glass transition) in

metallic glasses; but, it is the first to propose that the glass-to-glass transition could be a new and general method for producing ultrastable glasses, surpassing traditional PVD methods.

That is, what appears to be a simple shift in thinking could potentially lead to significant breakthroughs.

While Ediger achieved laboratory-timescale production of the ultrastable glasses, this study may pave the way for a new trend in the production of bulk ultrastable glasses. The shift from a surface-controlled process to a bulk transformation for generating ultrastable glasses is of considerable significance. Firstly, it means that, on a laboratory timescale (other than millions-year-long natural aging), the production of ultrastable glasses has finally broken through the two-dimensional limit. Secondly, anisotropy is no longer a concern since the glass-to-glass transition involves a monolithic structural adjustment. The successful fabrication of the three-dimensional uniform ultrastable glasses could provide an ideal candidate for studying the intrinsic structure of glasses and the nature of the glass transition, such as the question of whether microscopic heterogeneity is an inherent characteristic of glasses.

Additionally, the compositional dependence experiments are another highlight of this study. This experiment design is quite simple, yet the authors provide a self-consistent explanation from the perspective of the competition between the glass-to-glass transition and crystallization. This concise explanatory framework unifies previous explanations of glass-to-glass transition, advancing our understanding of the glass-to-glass transition; reveals the universality of the glass-to-glass transition, increasing our confidence for the widespread production of ultrastable glasses via glass-to-glass transition; and introduces a simple method to searching for more glasses with the glass-to-glass transition.

Based on the above significant progress in this work, I recommend publishing it in Nature Communications. I think it will prompt us to explore leveraging the transformation features of polyamorphous transition to address critical scientific challenges and technological applications.

Some further clarification on the following issues is necessary before publication:

(1) In contrast to vapor-deposited molecular glasses, which typically demonstrate enhanced kinetic and thermodynamic stability, the thermodynamic stability of vapor-deposited ultrastable MGs may indeed deteriorate further, as noted by Yu et al. *Adv. Mater* 2013, with their enthalpy surpassing that of liquid-cooled glass. Given this, I firmly believe that the benefits of achieving ultrastable glass through polyamorphous transition far outweigh those of the vapor deposition method, not to mention the size and preparation time advantages inherent in the sample.

(2) How does the author interpret the shift of the $G(r)$ peak towards a larger r in the glass state after heat treatment compared to the initial state, particularly the first nearest neighbor peak? Does a larger r indicate a looser structure? Does this conflict with the densification discussed later in the text? This structural alteration has also been observed in other MG systems undergoing polyamorphous transition phenomena [*Mater. Today*, 34, 66-77 (2020); *J. Phys. Chem. Lett.*, 11(16), 6718-6723 (2020); *Nat. Commun.*, 13(1), 2183 (2022); *J. Chem. Phys.*, 157(18), 184504 (2022)].

(3) Regarding Fig. 3, I suggest incorporating additional data points for components from two recent studies by Yang et al. [*J. Chem. Phys.*, 157(18), 184504 (2022); *Acta Mater.*, 266, 119701 (2024)]. Additionally, it is essential to note that the ΔT value for TiZrCuNiBe HEMG was measured at 1000 K/s, while for other components in Figure 3, it was calculated at a rate of approximately 20 K/min. Standardization is crucial as ΔT is influenced by the heating rate used. Furthermore, if the ΔT of the ultrastable glass increases compared to liquid-cooled glass, the X-axis of the graph should be treated similarly to how the δT_g axis of the prepared sample is handled. If there is no increase, it may be worth reconsidering the use of the ΔT axis. I posit that the supercooled liquid phase region in the glass represents the stability of the liquid rather than the glassy state, and this region's characteristics are influenced by composition.

Version 1:

Reviewer comments:

Reviewer #1

(Remarks to the Author)

For most of the issues raised by the reviewers, the authors have adjusted the manuscript appropriately. For one issue that I raised initially (repeated below), I do not think that the authors have done enough.

Comment from initial review: The comparison to PVD glasses is definitely relevant and important, but I think it requires a bit more explanation. For PVD glasses made from organic molecules, there is only one supercooled liquid – all the properties of the PVD glass are measured relative to a liquid-cooled glass of the same supercooled liquid. In the present manuscript, the heat-treated glass is a liquid-cooled glass, and its properties are compared to a liquid-cooled glass of a different supercooled liquid. From this perspective, the new glass that is prepared in this work is not so remarkable. Rather, it is the second supercooled liquid that is remarkable.

The authors have not addressed this adequately in the revised manuscript. I understand that their primary interest is the new glass that is formed – and I agree that the new glass is very interesting and has remarkable properties in comparison to the initial supercooled liquid. So, from a practical perspective, the authors succeeded in making a considerably denser and more kinetically stable glass, in comparison to the initial liquid-cooled glass. Here is the key difference: PVD prepares ultrastable glasses by circumventing the long time scales normally needed to age a glass, without changing the supercooled liquid. In the present work, a phase transition lowers the enthalpy of both the liquid and the glass; the new glass is very special compared to the initial glass, but in one sense, it is an ordinary liquid-cooled glass of a more stable liquid.

Since the authors mention that their new glasses provide an opportunity for advanced fundamental understanding, surely

they should provide a clear statement about this very fundamental point. I did not find such a statement in the main text of the revised manuscript (sorry if I missed it). They need to add 1-2 sentences that explains this clearly.

Mark Ediger

Reviewer #2

(Remarks to the Author)

the authors have addressed all my comments and concerns and I am looking forward to see this work discussed in the glass community.

Reviewer #3

(Remarks to the Author)

The authors clarified my previous concerns, I am pleased to recommend this version for publication.

Version 2:

Reviewer comments:

Reviewer #1

(Remarks to the Author)

Thank you for adding this important clarification.

The manuscript should be accepted for publication.

Mark Ediger

Point-by-point Response to the Reviewers' Comments

The reviewers' comments are in blue, and our responses in black.

Reviewer #1

This is a fascinating study of glass formation in a TiZrCuNiBe metallic alloy. The authors show that this system has a glass-to-glass transition (or liquid-to-liquid transition). They heat-treat the as-prepared glass to induce this transition, and then cool back to the glassy state. As a result, they prepared a glass that is high density and low enthalpy, relative to the initial rapidly-cooled glass. Interestingly, when the new glass is heated it yields a supercooled liquid that is quite stable against crystallization.

The authors make extensive comparison to ultrastable glasses prepared by physical vapor deposition (PVD). By some metrics, the bulk processed glasses prepared here are more stable than PVD glasses. This provides a very interesting point of comparison for this new work.

This manuscript is very clearly written and the experimental work appears to be of very high quality. I am excited to read about these new results and I am inclined to support publication in *Nature Communications*. I do have some important concerns, and I hope to be able to give my endorsement for publication after seeing how the authors respond to these concerns.

Reply: We are truly delighted and greatly encouraged to see your positive evaluation of our work and your support for its publication in *Nature Communications*. In response to your comments, we have outlined our understanding below, hoping that these replies will adequately address your concerns.

Comment 1

The comparison to PVD glasses is definitely relevant and important, but I think it requires a bit more explanation. For PVD glasses made from organic molecules, there is only one supercooled liquid – all the properties of the PVD glass are measured relative to a liquid-cooled glass of the same supercooled liquid. In the present

manuscript, the heat-treated glass is a liquid-cooled glass, and its properties are compared to a liquid-cooled glass of a different supercooled liquid. From this perspective, the new glass that is prepared in this work is not so remarkable. Rather, it is the second supercooled liquid that is remarkable.

Reply: Thank you for your insightful comment.

When a glass is heated, it undergoes a glass transition, characterized by a jump in specific heat, after which, it enters the supercooled liquid phase until crystallization occurs. For PVD glasses, it is generally believed that the deposited glasses and liquid-cooled glass share the same thermodynamic state in the supercooled liquid region[R1]. Based on this, the enthalpy and fictive temperature (T_f) can be determined by vertically shifting the enthalpy curves to overlap the enthalpies of supercooled liquids[R1,2].

In this work, the glass-to-glass transition occurred in the temperature range of 716 K to 781 K, above the T_g of the as-prepared glass ($T_{g,as-prepared} < T_{g,heat-treated} = 642 \text{ K} < 716 \text{ K}$). The as-prepared glass first transforms into a supercooled liquid (SCL1), followed by an exothermic event. Due to the exothermic event, the system could transform into another supercooled liquid (SCL2) with a different thermodynamic state. The ultrastable glass (heat-treated sample) is then derived directly from cooling the SCL2, resulting in significantly different structural and physical properties compared to the as-prepared glass.

We appreciate your suggestion on the emergence of SCL2 and have made some clarification in the manuscript, as it is indeed a remarkable aspect. However, our primary focus in this work is on the resulting ultrastable glass, rather than the detailed pathway. While the SCL2 is intriguing and could offer further research opportunities, in this work we have chosen to first concentrate on demonstrating this alternative approach to obtaining ultrastable glasses. Thank you once again for your thoughtful comments.

Comment 2

In the abstract and elsewhere, the authors propose that their approach is universal for metallic glasses. Yet they show several compositions where it does not work (without

adjusting the composition), therefore it is not universal for any given composition. I suggest that the authors drop the claim of universality and describe this as a “general” approach.

Reply: Thank you for your suggestion on the subtle difference between the two words, “universal” and “general”. This approach is indeed not universal for any given composition, and “general” is our original intention. We have replaced “universal” with “general” in the revised version.

Comment 3

In the abstract, the anisotropy of PVD glasses is described as a limitation. For some applications (organic electronics), anisotropy improves performance – so it does not seem fair to call this a limitation. It is a difference. Isotropy may be preferred in some applications.

Reply: Thank you for pointing out this potential issue. We have rephrased the sentence in the abstract from the original version “Here, we demonstrate an approach to accessing ultrastable glasses via the glass-to-glass transition, a bulk transformation that fundamentally overcomes the limitations of substrate support, size constraints, and anisotropy”, to the revised version “Here, we demonstrate an approach to accessing ultrastable glasses via the glass-to-glass transition, a bulk transformation that is inherently free from size constraints and anisotropy”.

Comment 4

About this sentence: “Ideal glass is expected to have a density approaching infinitely that of its crystalline counterpart...” I think you are completely incorrect on this point. For organic liquids, typical extrapolation of supercooled liquid properties show that the configurational entropy goes to zero (defining the ideal glass) at a temperature where the supercooled liquid is less dense than the crystal. If you are making a statement about metallic glasses in particular, please provide some references that support your claim.

Reply: Thank you for your comment. Based on the current consensus, the ideal glass is located at the lowest position on the potential energy landscape among all its glass states

and has the highest density[R3], but its density will not exceed that of its stable crystalline counterpart[R4]. It is experimentally challenging to obtain the density of the ideal glass but easy to obtain that of the crystalline counterpart. Therefore, in our work, we use the density of the crystalline counterpart as the upper limit of the glass density, in a conservative way, to estimate how far a glass state is from the ideal glass state. It is indeed inappropriate to say that the density of an ideal glass should approach infinitely that of its corresponding crystal. We have revised the sentence in the manuscript to “For the same composition, the density of the ideal glass should be the highest among all glassy states, but it is not supposed to exceed that of its stable crystalline counterpart, which sets an upper boundary of the densities of all glassy states”.

Comment 5

If I understand you correctly, the heat-treated glass is about 0.7% less dense than the crystallized sample. Please compare this number to other literature reports for metallic glasses. Is 0.7% unusually small compared to other metallic glasses? Is 3% (the difference between the ribbon sample and crystal) unusually large? Please include a careful comparison with the literature.

Reply: Thank you for your comment.

Table R1 summarizes the density differences ($\Delta\rho=(\rho_{\text{crys}}-\rho_{\text{amor}})/\rho_{\text{amor}}$) between some metallic glasses and their crystalline counterparts. The specific value of $\Delta\rho$ is influenced by the alloy composition and preparation process, but generally falls within the range of 1–4%. For glassy ribbons prepared by melt-spinning, it is widely accepted that their densities are 2–3% lower than those of crystalline counterparts[R5]. Therefore, the ~3% density difference between the liquid-cooled glass (as-prepared glassy ribbon) and its stable crystalline counterpart in this work is not unusually large. Meanwhile, the density of the heat-treated glass is only 0.7% lower than that of its crystalline counterpart, which is rather small. In this work, what we would like to highlight is the 2.3% increase in density resulting from the glass-to-glass transition. For contrast, relaxation as the typical path for tuning glass states usually results in a density increase of only 0.1–

0.5%[R6], making the density increase induced by the glass-to-glass transition far beyond what relaxation can achieve. Thank you for your comment again and we have included this discussion in Supplementary Note 2 and Supplementary Table 4.

Table R1 The density differences between metallic glasses and their crystalline counterparts.

Compositions of metallic glasses (at.%)		$\Delta\rho$ (%)	Ref.	
film	Cu-Zr	2.5–4.5	[R7]	
	Cu-Zr-Al	1–3	[R8]	
	Cu ₃₃ Zr ₆₇	1.9	[R9]	
	Cu ₅₉ Zr ₄₁	3.9	[R9]	
	Pd-Si	~2	[R10]	
ribbon	Ti ₄₅ Zr ₅ Ni ₃₅ Cu ₁₅	2.11	[R11]	
	Ti ₄₀ Zr ₁₀ Ni ₃₅ Cu ₁₅	1.74	[R11]	
	Ti ₃₅ Zr ₁₅ Ni ₃₅ Cu ₁₅	1.26	[R11]	
	Ti ₃₀ Zr ₂₀ Ni ₃₅ Cu ₁₅	1.08	[R11]	
	Ni _{59.5} Nb _{40.5}	3.36	[R12]	
	Pd-Cu-Si	1.87–2.30	[R13]	
	Pd-Si-Cu	1.87–2.23	[R13]	
	Pd-Ni-Fe-P	0.62–1.34	[R14]	
	bulk	Zr _{41.2} Ti _{13.8} Cu _{12.5} Ni ₁₀ Be _{22.5}	1.09	[R12]
		Zr ₅₇ Cu _{15.4} Ni _{12.6} Al ₁₀ Nb ₅	2.32	[R12]
Zr _{52.5} Cu _{17.9} Ni _{14.6} Al ₁₀ Ti ₅		2.23	[R12]	
Zr ₅₂ Ti ₅ Cu ₁₈ Ni ₁₅ Al ₁₀		2.58	[R15]	
Mg ₆₅ Cu _{7.5} Ni _{7.5} Zn ₅ Ag ₅ Y ₅ Gd ₅		0.98	[R16]	

Comment 6

Figure 2b seems not quite fair. For the PVD glasses, the density increase is calculated relative to a glass made by cooling the liquid at 1 K/min. For your sample, the density increase is calculated relative to the ribbon cooled sample (10⁴ K/s??). The difference in cooling rates unfairly makes your result relatively larger. This likely accounts for

only a part of the difference, but should be noted.

Reply: Thank you for your comment. When comparing the proportion of density increase, the reference glass we use was formed at a faster cooling rate, which indeed creates some unfairness. However, we still keep the figure mainly for two reasons:

(1) The glass-forming ability (GFA) of most metallic glasses is quite limited, making it challenging to form them at cooling rates of a few K/min. Consequently, even in studies on the preparation of ultrastable metallic glasses by the PVD method[R17,18], ribbon samples (i.e., metallic glasses obtained at a cooling rate of 10^4 – 10^6 K/s) are still used as references. Since the GFA of the TiZrCuNiBe system in this study is relatively weak, we chose to use the glassy ribbon sample obtained by a high cooling rate as the reference.

(2) The density of metallic glasses prepared at different cooling rates will vary, but the change is typically small. For example, as shown in Fig. R1, the density difference of a Pd₄₀Cu₃₀Ni₁₀P₂₀ metallic glass prepared at cooling rate from 500 K/s to 1.98 K/s is only about 0.16%[R19]. Furthermore, fitting the density data at different cooling rates of Pd₄₀Cu₃₀Ni₁₀P₂₀, results in an expression:

$$\rho = 0.0235 / R^{0.6551} + 9.2698 \quad (1)$$

with a coefficient of determination of 0.995, where ρ is the density and R is the cooling rate ($R > 1.98$ K/s to form a glass). Even if the cooling rate is very fast to approach infinity, the density would approach $\rho_{\infty} = 9.2698$ g/cm³, which is only 0.164% difference from the glass with the cooling rate at 1.98 K/s (9.2850 g/cm³). According to the fitting equation, density becomes even less sensitive to the cooling rate especially within the high cooling rate range.

Considering the above two facts, using a melt-spun sample as the reference is reasonable. We believe that the 2.3% density change is mainly caused by the glass-to-glass transition and is weakly related to the cooling rate. To make it clear for the readers, following your suggestion, we included the discussion above in Supplementary Note 2.

Figure Redacted

Fig. R1 Density as a function of the cooling rate for Pd₄₀Cu₃₀Ni₁₀P₂₀ BMG. The figure is from Ref. [R5,19].

Comment 7

The comparison of θ_K and θ_ρ cannot be quantitatively meaningful. See first major issue above.

Reply: Thank you for your comment.

As discussed in Comment 1, the calculation of θ_K relies on the acquisition of T_f , which requires a same supercooled liquid. In contrast, in this work, the as-prepared glass and the ultrastable glass originate from different supercooled liquids, making the θ_K parameter inapplicable here.

In this work, we propose θ_ρ as a substitute for θ_K to evaluate how far a glass state is from the ideal glass state. The density of the stable crystal is used as the upper boundary of densities, while the density of the melt-spinning glass (not sensitive to the cooling rate as addressed in Comment 6) serves as the lower boundary. For a glass to be evaluated, its density falls between these two limits, which can be used as an indicator of its state. The closer the density is to the crystalline state, the closer θ_ρ is to 1, indicating that the glass is in a state closer to the ideal glass.

Since both temperature and density are state parameters of a thermodynamic system, both θ_K and θ_ρ can be used to evaluate the state of a glass. In addition, both θ_K and θ_ρ contain an implicit assumption that the progression of different glass states toward the

ideal glass state is linearly related to their fictive temperature and density, respectively. Under this assumption, the comparison between θ_p and θ_K makes some sense, even though they are defined using different thermodynamic parameters.

Reviewer #2:

Ultrastable glasses through bulk phase transformation

This paper reports on using a glass-to-glass transition to fabricate ultrastable glasses. The research is exciting as it suggests a pathway to reach those “ultraold” glasses through a different mechanism, also different from the recently explored sputtering approach where surface mobility has been used to get into such deep enthalpic states. Glass to glass transitions have been reported in recent years (a lot of work has come out of Bill Johnson's group at Caltech which the authors should also acknowledge) but has not been associated with ultrastable glasses.

This is the main point of this paper. Clearly the authors provide a multitude of evidence for this claim and I would argue that all these evidences clearly point towards a highly in enthalpy reduced glass.

Reply: Thank you for your recognition of our work. As you rightly pointed out, linking the glass-to-glass transition with ultrastable glasses is the main point of our work. What we would like to illustrate is that we find another way for the preparation of ultrastable glasses, which is different from the traditional physical vapor deposition (PVD) method. Besides, as you mentioned, Bill Johnson's group has done a lot of related work, from which we draw a lot of inspiration. We highly appreciate their work and have cited the relevant articles in our manuscript[R20-23].

Comment 1

However, and this is the main comment I have is that if indeed the glass is an ultrastable glass that would mean it would continuously relate to a typical, less stable glass, through time scales of relaxation. In other words, how do we know that it is “the same glass” and not a different phase.

A different glass could be something like a different crystalline material (BCC vs FCC) and the same would be a supersaturated BCC which would reduce in saturation.

Fundamentally, and I strongly believe this work has potentially deep fundamental ramification (and not technological), such difference is important.

In other words, is the low enthalpy glass separated by a nucleation process or is it

continuous?

To test such continuous behaviour the author should anneal their glasses at different temperatures for longer than the relaxation time. The relaxation time can be estimated from similar glasses (~ 100 sec at T_g and shortening \sim by order of magnitude every 20 K increase).

This would allow some continuous sampling of properties such as density or modulus. I strongly encourage the authors to carry out such experiments as the outcome reveals if the glass is continuous connected or separated through a nucleation process.

Reply: Thank you very much for this insightful comment. Whether the glass-to-glass transition here occurs through a nucleation-growth process or a continuous transformation process is indeed an interesting question to explore.

As you have suggested, we annealed the as-prepared glass at each different temperature (740–790 K, with intervals of 10 K) for 5 min. For each of the annealed samples, we tested their DSC curves at a heating rate of 10 K/min and integrated the first exothermic peak (i.e., the peak denoting the glass-to-glass transition) to obtain the residual enthalpy after the annealing. The glass transition temperature (T_g) of each sample was measured using a flash DSC at a heating rate of 1000 K s⁻¹. Then, the modulus and hardness of the annealed samples were measured using the nanoindentation tests. The residual enthalpy, T_g , modulus, and hardness of the annealed samples versus the annealing temperatures were plotted in Fig. R2.

Fig. R2 Continuous evolution of glass properties with progressive GGT. The as-prepared glass was annealed at 6 different temperatures from 740–790 K to gradually induce the GGT and monitor the resulting property changes. **a** The residual enthalpy of the first exothermic peak (GGT peak) and the

glass transition temperature (T_g) of the annealed samples. **b** The modulus and hardness of the annealed samples.

As the annealing temperature increases, the glass-to-glass transition gradually occurs, manifesting as a gradual decrease in the exothermic heat of the first exothermic peak of the annealed samples in the DSC. At the same time, we observed that (1) there is only one T_g for each sample; (2) as the annealing temperature increases, the T_g of the corresponding sample increases (see Fig. R2). Only one glass phase exists during the glass-to-glass transition, ruling out the possibility of the nucleation of a new glass phase, during which two glass phases would co-exist. On the contrary, the low-energy glass continuously relates to a less stable glass, which is confirmed by progressively increased T_g (see Fig. R2a), as well as progressively increased modulus and hardness (see Fig. R2b).

Based on the results of the enthalpy, modulus, hardness, and especially the measurement of T_g , we conclude that the glass is continuously connected rather than separated by a nucleation process during the glass-to-glass transition, at least in this work. Thank you again for this insightful question. We will address it with more experiments and discussions in another paper, and thus we would not include the data of Fig. R2 in this work.

Comment 2

the authors should cite Bill Johnson work on glass to glass transition.

Reply: Thank you for your suggestion. We highly appreciate the work of Bill Johnson's research group and have already cited the relevant literature [R20-23] in the revised version.

Comment 3

I found the mentioning of technological relevance of this work far stretched and unnecessary. To me, this is pure science and potentially very exciting as it challenges our fundamental understanding of relaxation phenomena in glasses. Not every work has

to be technologically relevant and by mentioning both, the scientific importance is diminished.

Reply: Thank you for your comment. The main value of this work lies in its scientific significance. Overemphasizing the technical value could, in fact, diminish the scientific value. We appreciate your suggestion and have removed some technically related expressions from the revised manuscript.

Comment 4

Further, ultrastable glasses are considered to have a very low fictive temperature. Fictive temperature has been directly correlated with mechanical properties (see [J. Ketkaew, Mechanical glass transition revealed by the fracture toughness of metallic glasses, *Nat Commun* 9 (2018).] [S.V. Ketov, Rejuvenation of metallic glasses by non-affine thermal strain, *Nature* 524(7564) (2015) 200-+.] [S. Sohn, A framework for plasticity in metallic glasses, *Materialia* 31 (2023) 101876.] and the author should mention this correlation (structure change through T_f results in mechanical property change and acknowledge this work.

Reply: Thank you for your comment. As you mentioned, an ultrastable glass should exhibit a very low fictive temperature, which is considered one of the signs of its thermodynamic stability. However, in this work, due to the presence of an exothermic peak related to the glass-to-glass transition in the DSC curve of the as-prepared glass, determining the fictive temperature is quite challenging. Therefore, we did not mention the fictive temperature in the manuscript, but instead used the reduction in enthalpy and the increase in density to illustrate its thermodynamic stability. We understand the correlation between the fictive temperature and mechanical properties, but since the fictive temperature cannot be determined experimentally, we are unable to explore this correlation. Thank you for your suggestion again.

Comment 5

the fact that you can fabricate the ultrastable glass “free from substrate” is trivial compare to the fact that it happens in bulk so quickly which is a very big finding.

Reply: We are grateful for your comment. We have removed the phrase “limitations of substrate support” from the abstract to highlight the key point of the paper, i.e., a rapid bulk process for fabricating ultrastable glasses. The sentence in the abstract “Here, we demonstrate an approach to accessing ultrastable glasses via the glass-to-glass transition, a bulk transformation that fundamentally overcomes the limitations of substrate support, size constraints, and anisotropy” was revised to “Here, we demonstrate an approach to accessing ultrastable glasses via the glass-to-glass transition, a bulk transformation that is inherently free from size constraints and anisotropy”.

Comment 6

line81: “proceeding 75% toward the ideal glass state.” How do you know the 100% ideal glass?

Reply: Thank you for your question. The ideal glass is located at the lowest position on the potential energy landscape among all its glass states. To estimate how far a glass state is from the ideal glass state, it is necessary to identify a parameter or indicator that represents the ideal glass state. However, the 100% ideal glass is an “ideal” state that no properties can be experimentally accessed so far. Fortunately, we have known that the density of the ideal glass is expected to be the highest among all glassy states[R3], which will be very close to but not exceed that of its stable crystalline counterpart[R4]. So, the density of the stable crystalline counterpart can be viewed as an upper boundary for the densities of all glassy states. To make a conservative estimate, we assume the density of the crystalline counterpart as the density of the ideal glass. The liquid-cooled glass, on the other hand, has a density close to the lowest, which could be regarded as the lower boundary of the densities. Then we defined a parameter θ_ρ in the previous version as a merit of how far a glass is from the ideal glass:

$$\theta_\rho = \frac{\rho - \rho_{\text{LCG}}}{\rho_{\text{crys}} - \rho_{\text{LCG}}} \quad (2)$$

where ρ , ρ_{LCG} , and ρ_{crys} represent the densities of the glass to be estimated, the liquid-cooled glass, and the stable crystalline counterpart, respectively. For a glass, θ_ρ ranges

between 0 and 1, where a larger value indicates a state closer to the ideal glass state. For the heat-treated TiZrCuNiBe glass, θ_p equals 0.75, indicating that it has proceeded 75% toward the ideal glass state. Of course, it should be noted that using θ_p to evaluate how far a glass state is from the ideal glass state is somewhat underestimated, as the density of the ideal glass is actually less but not strictly equal to that of its crystalline counterpart.

Reviewer #3:

Although the PVD method has been used to prepare amorphous thin films for a long time, it was not until 2007 that Ediger successfully generated ultrastable glasses just by elevating the substrate temperatures.

Similarly, this work is not the first report of the thermally induced polyamorphous transition (glass-to-glass transition) in metallic glasses; but, it is the first to propose that the glass-to-glass transition could be a new and general method for producing ultrastable glasses, surpassing traditional PVD methods.

That is, what appears to be a simple shift in thinking could potentially lead to significant breakthroughs.

While Ediger achieved laboratory-timescale production of the ultrastable glasses, this study may pave the way for a new trend in the production of bulk ultrastable glasses. The shift from a surface-controlled process to a bulk transformation for generating ultrastable glasses is of considerable significance. Firstly, it means that, on a laboratory timescale (other than millions-year-long natural aging), the production of ultrastable glasses has finally broken through the two-dimensional limit. Secondly, anisotropy is no longer a concern since the glass-to-glass transition involves a monolithic structural adjustment. The successful fabrication of the three-dimensional uniform ultrastable glasses could provide an ideal candidate for studying the intrinsic structure of glasses and the nature of the glass transition, such as the question of whether microscopic heterogeneity is an inherent characteristic of glasses.

Additionally, the compositional dependence experiments are another highlight of this study. This experiment design is quite simple, yet the authors provide a self-consistent explanation from the perspective of the competition between the glass-to-glass transition and crystallization. This concise explanatory framework unifies previous explanations of glass-to-glass transition, advancing our understanding of the glass-to-glass transition; reveals the universality of the glass-to-glass transition, increasing our confidence for the widespread production of ultrastable glasses via glass-to-glass transition; and introduces a simple method to searching for more glasses with the glass-to-glass transition.

Based on the above significant progress in this work, I recommend publishing it in *Nature Communications*. I think it will prompt us to explore leveraging the transformation features of polyamorphous transition to address critical scientific challenges and technological applications.

Reply: We are very grateful for your praise of our work. Your generous compliments are a great encouragement to us, and we will continue to make this work more comprehensive and in-depth.

Comment 1

In contrast to vapor-deposited molecular glasses, which typically demonstrate enhanced kinetic and thermodynamic stability, the thermodynamic stability of vapor-deposited ultrastable MGs may indeed deteriorate further, as noted by Yu et al. *Adv. Mater* 2013, with their enthalpy surpassing that of liquid-cooled glass. Given this, I firmly believe that the benefits of achieving ultrastable glass through polyamorphous transition far outweigh those of the vapor deposition method, not to mention the size and preparation time advantages inherent in the sample.

Reply: Thank you for the comment. We completely agree with you. In 2013, Yu[R17] first applied the physical vapor deposition (PVD) method to prepare ultrastable metallic glasses. An interesting discovery was that, unlike what was observed in molecular glasses, despite higher kinetic stability, the deposited metallic glass exhibits worse thermodynamic stability. In our work, the ultrastable glass induced by the glass-to-glass transition is obviously in a thermodynamic lower-energy state. In this regard, we agree with what you've said: "The benefits of achieving ultrastable glass through polyamorphous transition far outweigh those of the vapor deposition method", at least in the metallic glasses.

However, we consider that making such a comparison solely within the field of metallic glasses is unfair to the PVD method. After all, most molecular glasses prepared by the PVD method are thermodynamically more stable[R24]. Therefore, we chose not to emphasize this point in this work. Thank you for your suggestion again.

Comment 2

How does the author interpret the shift of the $G(r)$ peak towards a larger r in the glass state after heat treatment compared to the initial state, particularly the first nearest neighbor peak? Does a larger r indicate a looser structure? Does this conflict with the densification discussed later in the text? This structural alteration has also been observed in other MG systems undergoing polyamorphous transition phenomena [*Mater. Today*, 34, 66-77 (2020); *J. Phys. Chem. Lett.*, 11(16), 6718-6723 (2020); *Nat. Commun.*, 13(1), 2183 (2022); *J. Chem. Phys.*, 157(18), 184504 (2022)].

Reply: Thank you for your comment. It is known that crystalline materials have long-range translational symmetry, and under an affine deformation, the shift of any crystalline plane can be used to estimate the overall density change. However, metallic glasses do not have any translation symmetry and the structural transformation is always non-affine. The shift of any atom-shell, especially the first peak in $G(r)$, cannot be directly related to the density change. For the ultrastable glass, it is noticed in Fig. 1d that the decay of the $G(r)$ peaks significantly slows down, and the peak intensity after the third peak of the $G(r)$ for the ultrastable glass are significantly higher than those of the liquid-cooled glass. This indicates more atoms are squeezed into these atom shells and therefore the density of the ultrastable glass will increase. A qualitative method for estimating the density change of metallic glasses based on diffraction data is to use the first peak of $S(Q)$ but not $G(r)$, as the first peak of $S(Q)$ contains not only the short-range information but also the medium-range to long-range information. The first peak of $S(Q)$ of the ultrastable glass shifts towards a larger Q direction compared to that of the liquid-cooled sample (shifts by $\sim 0.4\%$, see in Supplementary Table 1), indicating that the ultrastable glass is denser than the liquid-cooled glass. Therefore, the diffraction data is consistent but does not conflict with our density measurement results.

Comment 3

Regarding Fig. 3, I suggest incorporating additional data points for components from two recent studies by Yang et al. [*J. Chem. Phys.*, 157(18), 184504 (2022); *Acta Mater.*, 266, 119701 (2024)].

Reply: Thank you for your suggestion.

In [*J. Chem. Phys.*, 157, 184504 (2022)], a $\text{La}_{65}\text{Co}_{25}\text{Al}_{10}$ MG was investigated and two methods, conventional DSC at a heating rate of 40 K/min and TMDSC (Temperature Modulated Differential Scanning Calorimetry) at a heating rate of 3 K/min, were used to detect the increase in T_g induced by the glass-to-glass transition. In conventional DSC testing, the liquid-cooled glass has a significant structural relaxation exothermic peak, which makes the determination of T_g questionable[R25]. Therefore, we chose to use the results from TMDSC instead of the conventional DSC results. This data point was added to Fig. 3 and the updated Fig. 3 was presented below named Fig. R3.

In [*Acta Mater.*, 266, 119701 (2024)], after the glass-to-glass transition of a ZrTiHfCoNi amorphous alloy, the T_g measured by flash DSC was increased by 60 K[R26]. However, the heat-treated sample did not exhibit a clear T_g in the conventional DSC test, thus making it impossible to calculate ΔT . As a result, this data point cannot be presented in Fig. 3. However, we have added Supplementary Table 5 in the Supplementary Information file, summarizing the δT_g and ΔT data for ultrastable metallic glasses, with this data included.

Fig. R3 Kinetic stability of the ultrastable TiZrCuNiBe glass induced by GGT. δT_g represents the increment of the onset glass transition temperature ($\delta T_g = (T_{g,L} - T_{g,H}) / T_{g,H}$, subscripts 'L' and 'H' mean low- and high-energy state, respectively); ΔT represents the width of the supercooled liquid region for the low-energy glass.

Comment 4

Additionally, it is essential to note that the ΔT value for TiZrCuNiBe HEMG was measured at 1000 K/s, while for other components in Figure 3, it was calculated at a rate of approximately 20 K/min. Standardization is crucial as ΔT is influenced by the heating rate used.

Reply: The ΔT value of the heat-treated TiZrCuNiBe glass was calculated using the data from conventional DSC test at a rate of 10 K/min instead of 1000 K/s. All ΔT values in Fig. 3 were calculated using the data from conventional DSC tests at a rate of 10 or 20 K/min. A heating rate of 10 K/min or 20 K/min has a negligible effect on the obtained ΔT values, so the comparison of ΔT in Fig. 3 is reasonable. We have added this discussion to the figure caption to avoid possible confusion.

Comment 5

Furthermore, if the ΔT of the ultrastable glass increases compared to liquid-cooled glass, the X-axis of the graph should be treated similarly to how the δT_g axis of the prepared sample is handled. If there is no increase, it may be worth reconsidering the use of the ΔT axis. I posit that the supercooled liquid phase region in the glass represents the stability of the liquid rather than the glassy state, and this region's characteristics are influenced by composition.

Reply: Thank you for your comment. We believe that comparing the ΔT of the liquid-cooled glass and ultrastable glass does not have much significance. The width of the supercooled liquid region ΔT refers to the difference between the onset crystallization temperature T_x and the glass transition temperature T_g . For the liquid-cooled glass, its onset crystallization temperature is the onset temperature of the second exothermic peak. For this reason, the liquid-cooled glass naturally has a much wider supercooled liquid region than that of the ultrastable glass, but this does not mean the supercooled liquid of the liquid-cooled glass is more stable than that of the ultrastable glass. On the contrary, the liquid-cooled glass is unstable in its supercooled liquid region as a glass-to-glass transition will occur. Therefore, it may be unreasonable to calculate the ΔT of

the liquid-cooled glass and compare it with that of the ultrastable glass.

In fact, assessing the ΔT of an ultrastable glass is of significance, as it accurately reflects the stability of its supercooled liquid. Indeed, as you have mentioned, ΔT does not directly reflect the stability of the glass but rather the stability of the supercooled liquid. Nevertheless, ΔT is related to the onset crystallization temperature T_x ($T_x = T_g + \Delta T$), and thus, along with T_g , ΔT can serve as an indicator of the glass's resistance to crystallization. Considering the reasons mentioned above, we believe that using ΔT as an evaluation indicator is justified.

References

- [R1] Kearns KL, Swallen SF, Ediger MD. Hiking down the Energy Landscape: Progress Toward the Kauzmann Temperature via Vapor Deposition. *J. Phys. Chem. B* **112**, 4934-4942, (2008). <https://doi.org/10.1021/jp7113384>.
- [R2] Swallen SF *et al.* Organic Glasses with Exceptional Thermodynamic and Kinetic Stability. *Science* **315**, 353-356, (2007). <https://doi.org/10.1126/science.1135795>.
- [R3] Beasley MS, Bishop C, Kasting BJ, Ediger MD. Vapor-Deposited Ethylbenzene Glasses Approach "Ideal Glass" Density. *J. Phys. Chem. Lett.* **10**, 4069-4075, (2019). <https://doi.org/10.1021/acs.jpcclett.9b01508>.
- [R4] Cui X-Y, Ringer SP, Wang G, Stachurski ZH. What should the density of amorphous solids be? *J. Chem. Phys.* **151**, (2019). <https://doi.org/10.1063/1.5113733>.
- [R5] Suryanarayana C, Inoue A. Bulk Metallic Glasses.
- [R6] Goncharova EV, Konchakov RA, Makarov AS, Kobelev NP, Khonik VA. On the nature of density changes upon structural relaxation and crystallization of metallic glasses. *J. Non-Cryst. Solids* **471**, 396-399, (2017). <https://doi.org/10.1016/j.jnoncrysol.2017.06.024>.
- [R7] Li Y, Guo Q, Kalb JA, Thompson CV. Matching Glass-Forming Ability with the Density of the Amorphous Phase. *Science* **322**, 1816-1819, (2008). <https://doi.org/10.1126/science.1163062>.
- [R8] Guo Q *et al.* Density change upon crystallization of amorphous Zr-Cu-Al thin films. *Acta Mater.* **58**, 3633-3641, (2010). <https://doi.org/10.1016/j.actamat.2010.02.033>.
- [R9] Altounian Z, Guo-hua T, Strom-Olsen JO. Crystallization characteristics of Cu-Zr metallic glasses from $\text{Cu}_{70}\text{Zr}_{30}$ to $\text{Cu}_{25}\text{Zr}_{75}$. *J. Appl. Phys.* **53**, 4755-4760, (1982). <https://doi.org/10.1063/1.331304>.
- [R10] Masumoto T, Kimura H, Inoue A, Waseda Y. Structural Stability of Amorphous Metals. *Materials Science and Engineering* **23**, 141-144, (1976). [https://doi.org/10.1016/0025-5416\(76\)90183-X](https://doi.org/10.1016/0025-5416(76)90183-X).
- [R11] Kim W-C *et al.* Enhancement of superelastic property in Ti-Zr-Ni-Cu alloy by using glass alloy precursor with high glass forming ability. *Acta Mater.* **173**, 130-141, (2019). <https://doi.org/10.1016/j.actamat.2019.04.062>.
- [R12] Mukherjee S, Schroers J, Zhou Z, Johnson WL, Rhim WK. Viscosity and specific volume of bulk metallic glass-forming alloys and their correlation with glass forming ability. *Acta Mater.* **52**, 3689-3695, (2004). <https://doi.org/10.1016/j.actamat.2004.04.023>.
- [R13] Chen HS, Park BK. Role of chemical bonding in metallic glasses. *Acta Metall.* **21**, 395-400,

- (1973). [https://doi.org/10.1016/0001-6160\(73\)90196-X](https://doi.org/10.1016/0001-6160(73)90196-X).
- [R14] Shen TD, He Y, Schwarz RB. Bulk amorphous Pd-Ni-Fe-P alloys: Preparation and characterization. *J. Mater. Res.* **14**, 2107-2115, (1999). <https://doi.org/10.1557/JMR.1999.0285>.
- [R15] Mattern N *et al.* Thermal behavior and glass transition of Zr-based bulk metallic glasses. *Materials Science and Engineering: A* **375-377**, 351-354, (2004). <https://doi.org/10.1016/j.msea.2003.10.125>.
- [R16] Park ES, Kim DH. Formation of Mg - Cu - Ni - Ag - Zn - Y - Gd Bulk Glassy Alloy by Casting into Cone-shaped Copper Mold in Air Atmosphere. *J. Mater. Res.* **20**, 1465-1469, (2005). <https://doi.org/10.1557/jmr.2005.0181>.
- [R17] Yu HB, Luo Y, Samwer K. Ultrastable metallic glass. *Adv. Mater.* **25**, 5904-5908, (2013). <https://doi.org/10.1002/adma.201302700>.
- [R18] Luo P *et al.* Ultrastable metallic glasses formed on cold substrates. *Nat. Commun.* **9**, 1389, (2018). <https://doi.org/10.1038/s41467-018-03656-4>.
- [R19] Hu X, Ng SC, Feng YP, Li Y. Cooling-rate dependence of the density of Pd₄₀Ni₁₀Cu₃₀P₂₀ bulk metallic glass. *Physical Review B* **64**, (2001). <https://doi.org/10.1103/PhysRevB.64.172201>.
- [R20] An Q, Johnson WL, Samwer K, Corona SL, Goddard WA. Formation of two glass phases in binary Cu-Ag liquid. *Acta Mater.* **195**, 274-281, (2020). <https://doi.org/10.1016/j.actamat.2020.05.060>.
- [R21] An Q, Johnson WL, Samwer K, Corona SL, Goddard WA, III. First-Order Phase Transition in Liquid Ag to the Heterogeneous G-Phase. *J. Phys. Chem. Lett.* **11**, 632-645, (2020). <https://doi.org/10.1021/acs.jpcclett.9b03699>.
- [R22] An Q, Johnson WL, Samwer K, Corona SL, Goddard WA. The first order L-G phase transition in liquid Ag and Ag-Cu alloys is driven by deviatoric strain. *Scr. Mater.* **194**, (2021). <https://doi.org/10.1016/j.scriptamat.2020.113695>.
- [R23] An Q *et al.* The L-G phase transition in binary Cu-Zr metallic liquids. *PCCP* **24**, 497-506, (2021). <https://doi.org/10.1039/d1cp04157f>.
- [R24] Ediger MD. Perspective: Highly stable vapor-deposited glasses. *J. Chem. Phys.* **147**, 210901, (2017). <https://doi.org/10.1063/1.5006265>.
- [R25] Yang Q *et al.* Structural length-scale of beta relaxation in metallic glass. *J. Chem. Phys.* **157**, 184504, (2022). <https://doi.org/10.1063/5.0123202>.
- [R26] Yang Q, Yang X-M, Zhang T, Liu X-W, Yu H-B. Structure and entropy control of polyamorphous transition in high-entropy metallic glasses. *Acta Mater.* **266**, (2024). <https://doi.org/10.1016/j.actamat.2024.119701>.

Point-by-point Response to the Reviewers' Comments

The reviewers' comments are in blue, and our responses in black.

Reviewer #1

For most of the issues raised by the reviewers, the authors have adjusted the manuscript appropriately. For one issue that I raised initially (repeated below), I do not think that the authors have done enough.

Comment from initial review: The comparison to PVD glasses is definitely relevant and important, but I think it requires a bit more explanation. For PVD glasses made from organic molecules, there is only one supercooled liquid – all the properties of the PVD glass are measured relative to a liquid-cooled glass of the same supercooled liquid. In the present manuscript, the heat-treated glass is a liquid-cooled glass, and its properties are compared to a liquid-cooled glass of a different supercooled liquid. From this perspective, the new glass that is prepared in this work is not so remarkable. Rather, it is the second supercooled liquid that is remarkable.

The authors have not addressed this adequately in the revised manuscript. I understand that their primary interest is the new glass that is formed – and I agree that the new glass is very interesting and has remarkable properties in comparison to the initial supercooled liquid. So, from a practical perspective, the authors succeeded in making a considerably denser and more kinetically stable glass, in comparison to the initial liquid-cooled glass. Here is the key difference: PVD prepares ultrastable glasses by circumventing the long time scales normally needed to age a glass, without changing the supercooled liquid. In the present work, a phase transition lowers the enthalpy of both the liquid and the glass; the new glass is very special compared to the initial glass, but in one sense, it is an ordinary liquid-cooled glass of a more stable liquid.

Since the authors mention that their new glasses provide an opportunity for advanced fundamental understanding, surely they should provide a clear statement about this very fundamental point. I did not find such a statement in the main text of the revised manuscript (sorry if I missed it). They need to add 1-2 sentences that explains this

clearly.

Reply: Thank you for your comments.

In response to your comment, we provided a brief explanation regarding the emergence of the second supercooled liquid in the previous revised manuscript: “In this study, however, due to the exothermic event in the supercooled liquid, another supercooled liquid with distinct thermodynamic state could form”. However, as you pointed out, this was not sufficiently clear in distinguishing between the PVD method and the GGT method.

To make it clearer, we have added the following explanation in the penultimate paragraph of the Discussion section: “Both the PVD and the GGT method successfully circumvent this long timescale, but through different pathways. The former leverages rapid surface diffusion, allowing molecules/atoms to quickly rearrange into lower-energy configurations, without altering the supercooled liquid. The latter utilizes a phase transformation to produce a lower-energy supercooled liquid, which is then frozen to RT to form a more stable glass”.

As two approaches for preparing ultrastable glasses on the laboratory timescale, the PVD and GGT methods work through distinct mechanisms, each offering its own advantages. We hope the revised manuscript clearly conveys the differences between these two processes. Thank you once again for your essential input.